# SPEAR🚀 : Receiver-to-Receiver Acoustic Neural Warping Field

## Abstract

We present *SPEAR*, a continuous receiver-to-receiver acoustic neural warping field for spatial acoustic effects prediction in an acoustic 3D space with a single stationary audio source. Unlike traditional source-to-receiver modelling methods that require prior space acoustic properties knowledge to rigorously model audio propagation from source to receiver, we propose to predict by warping the spatial acoustic effects from one reference receiver position to another target receiver position, so that the warped audio essentially accommodates all spatial acoustic effects belonging to the target position. *SPEAR* can be trained in a data much more readily accessible manner, in which we simply ask two robots to independently record spatial audio at different positions. We further theoretically prove the universal existence of the warping field if and only if one audio source presents. Three physical principles are incorporated to guide *SPEAR* network design, leading to the learned warping field physically meaningful. We demonstrate *SPEAR* superiority in receiver-to-receiver warping field prediction through detailed experiments on both synthetic, photo-realistic and real-world dataset.

## 1    Introduction

In an enclosed acoustic 3D space where a stationary sound source keeps emitting spatial audio, the primary objective is to precisely delineate the spatial acoustic effects for any given receiver position. These spatial acoustic effects typically encompass reverberation, loudness variation and resonance. Achieving high-fidelity and authentic spatial acoustic effect modelling is pivotal for delivering a truly immersive 3D acoustic experience that seamlessly integrates with the 3D room scene. Consequently, such modelling techniques have a wide range of applications in auditory AR/VR techniques (Verron et al., 2010; Hyodo et al., 2021; Broderick et al., 2018), audio-inclusive robot tasks (Evers et al., 2016) and reconstruction endeavors (Zhong et al., 2022; Chen et al., 2021).

To model the spatial acoustic effects, most prior methods (Bilbao & Hamilton, 2017; Savioja & Svensson, 2015; Allen & Berkley, 1979; Krokstad et al., 1968; Pietrzyk, 1998; Kleiner et al., 1993; Botteldoore, 1995; Hodgson & Nosal, 2006; Luo et al., 2022) follow the source-to-receiver pipeline to explicitly model sound propagation process from source to receiver, where the overall behavioural change along the propagation path is usually described by room impulse response (RIR). As the spatial audio propagates in a complex way encompassing diffraction, reflection and absorption, the resulting RIR is highly non-smooth and lengthy in data points. Classic methods, either wave-based (Bilbao & Hamilton, 2017; Pietrzyk, 1998; Kleiner et al., 1993; Botteldoore, 1995) and geometry-based modelling (Savioja & Svensson, 2015), require massive prior knowledge of the 3D space's acoustic properties such as source position, space geometric layout and constructional material to precisely simulate the propagation process for a given source-receiver pair. However, accessing such prior knowledge poses a formidable challenge in reality and the whole computation is inextricably inefficient. Some recent work (*e.g.*, NAF (Luo et al., 2022)) aim to alleviate this computational burden by learning a continuous acoustic neural field. Nonetheless, training such a continuous field necessitates vast RIR data which is exceedingly difficult to collect in real scenarios.

In this work, we instead propose to predict spatial acoustic effects from receiver-to-receiver perspective. Our framework, termed *SPEAR*, relies on neither RIR data nor prior space acoustic properties that are difficult to obtain and required by existing source-to-receiver based methods (Luo et al., 2022; Savioja & Svensson, 2015; Bilbao & Hamilton, 2017), but instead simply require much more

Figure 1: **SPEAR Motivation**: A stationary audio source is emitting audio in 3D space. Requiring neither source position nor 3D space acoustic properties, *SPEAR* simply requires two microphones to actively record the spatial audio independently at discrete positions. During training, *SPEAR* takes as input a pair of receiver positions and outputs a warping field potentially warping the recorded audio on reference position to target position. Minimizing the discrepancy between the warped audio and recorded audio enforces *SPEAR* to acoustically characterise the 3D space from receiver-to-receiver perspective. The learned *SPEAR* is capable of predicting spatial acoustic effects at arbitrary positions.

readily accessible data – the receiver recorded audio at discrete positions. Our observation is that directly carrying a receiver (can ask robot or human to hold the receiver) to record audio at different positions is much readily executable than measuring RIR data and space acoustic properties. Since receiver recorded audio naturally encodes the spatial acoustic effects at its position, analyzing the received audio can help to acoustically characterize the 3D space, and further predict the spatial acoustic effects for any given novel receiver position.

As is shown in Fig. 1, to obtain the training data, we simply require receivers to record the audio in the 3D space independently. At each discrete position pair, the two receivers are temporally synchronized to record the same audio content and their respective positions are recorded as well. *SPEAR* then learns a continuous receiver-to-receiver acoustic neural warping field that takes as input two receivers' position and outputs a neural warping field warping the spatial acoustic effects from one reference position to the other target position. With the learned *SPEAR*, we can warp the audio recorded at one arbitrary reference position to another arbitrary target position so that the warped audio fully accommodates the spatial acoustic effects at the target position. *SPEAR* has huge potential in various robotic tasks, such as audio-involved robot relocalization and manipulation.

We further theoretically prove the universal existence of the receiver-to-receiver warping field if and only if one stationary audio source presents in the 3D space, then we introduce three main physical principles underpinned by linear time-invariant (LTI) 3D acoustic space that guide *SPEAR* neural network design: *Globality*, *Order Awareness* and *Audio-Content Agnostic*. We adopt the Transformer (Vaswani et al., 2017) architecture to predict the warping field in frequency domain, where the lengthy warping field is divided into small and non-overlapping patches. Each token is responsible for predicting a patch. We run experiments on both synthetic, photo-realistic and real-world datasets to show the superity of *SPEAR*. In summary, we make three main contributions:

1. We propose *SPEAR*, a novel receiver-to-receiver spatial acoustic effects prediction framework. Unlike existing source-to-receiver modelling methods requiring extensive prior space acoustic properties knowledge, *SPEAR* can be efficiently trained in a data more readily accessible manner.

2. We theoretically prove the universal existence of receiver-to-receiver neural warping field if and only if one stationary audio source presents in the 3D space. *SPEAR* network design is based on three acoustic physical principles, so that the whole neural network is physically meaningful.

3. We demonstrate *SPEAR* superiority on both synthetic data, photo-realistic and real-world dataset.

## 2 RELATED WORK

**Spatial Acoustic Effects Modelling.** Classic methods tend to numerically compute room impulse response (RIR). They can be divided into two main categories: wave-based (Bilbao & Hamilton, 2017; Pietrzyk, 1998; Kleiner et al., 1993; Botteldoore, 1995) and geometry-based (aka geometrical acoustics) (Savioja & Svensson, 2015; Krokstad et al., 1968; Allen & Berkley, 1979; Funkhouser et al., 2003; Hodgson & Nosal, 2006; Nosal et al., 2004). They require extensive prior knowledge of the 3D space acoustic properties such as the audio source position (He et al., 2021), room geometry and furniture arrangement to derive RIR. Moreover, they are computationally expensive and the

whole computation needs to be resumed once either the source or receiver position gets changed. Our proposed *SPEAR* circumvents the dependency on extensive prior knowledge and learns the spatial acoustic effects in a data more readily accessible manner. The advent of deep neural networks has inspired recent work (Ratnarajah et al., 2022; Tang et al., 2020; Ratnarajah et al., 2021b;a; De Sena et al., 2015; Luo et al., 2022; Richard et al., 2022; Majumder et al., 2022; Ratnarajah et al., 2023; 2024) to focus on learning RIR with deep neural networks. While showing promising performance, they still require massive RIR data or even crossdomain visual data (Ratnarajah et al., 2024) to train their model, which in reality are difficult to collect. While all of those methods fall into source-to-receiver estimation, *SPEAR* infers spatial acoustic effects from a receiver-to-receiver perspective which naturally brings several advantages over existing methods.

**Neural Implicit Representation.** Implicit representation learning has received lots of attention in recent years, especially in computer vision community (Mildenhall et al., 2020; Hedman et al., 2021; Xu et al., 2021; Su et al., 2021; Yu et al., 2021). They model static or dynamic visual scenes by optimizing an implicit neural radiance field in order to render photo-realistic novel views. Some recent work (*e.g.*, NAF (Luo et al., 2022), FewShotRIR (Majumder et al., 2022)) propose to learn implicit neural acoustic fields from source-receiver pairs or audio-visual cues. *SPEAR* also learns a spatial continuous neural implicit representation for spatial acoustic effects prediction.

**Audio Synthesis**. Estimating the audio for at a novel position partially relates to audio synthesis (Oord et al., 2016; Donahue et al., 2019; Zuiderveld et al., 2021; Richard et al., 2021; Engel et al., 2019; Tan et al., 2012; Clarke et al., 2021; Prenger et al., 2019). WaveNet (Oord et al., 2016) learns to predict future sound waveform based on previously heard sound waveform. WaveGAN (Donahue et al., 2019) and GANSynth (Engel et al., 2019) adopt generative adversarial network (GAN (Goodfellow et al., 2014)) to learn to generate audio. Our framework *SPEAR* differs from audio synthesis as it focuses on spatial acoustic effects modelling.

**Time-series Prediction.** Predicting warping field in either time or frequency domain partially relates to time-series prediction. Existing deep neural network based time-series prediction methods can be divided into four main categories: Convolutional Neural Networks (CNNs) based (Zheng et al., 2014; Yang et al., 2015; Wang et al., 2017; Foumani et al., 2021), Recurrent Neural Networks based (Dennis et al., 2019; Tang et al., 2016; Sutskever et al., 2014), Graph Neural Networks based (GNNs) (Covert et al., 2019; Jia et al., 2020; Ma et al., 2021; Tang et al., 2022; Zhang et al., 2022) and Transformer based (Song et al., 2018; Jin & Chen, 2021; Liu et al., 2021) methods. However, warping field prediction in *SPEAR* exhibits no causality and it is directly predicted from receiver positions.

## 3 RECEIVER-TO-RECEIVER ACOUSTIC NEURAL WARPING FIELD

### 3.1 PROBLEM FORMULATION

In a linear time-invariant (LTI) enclosed 3D space $\mathcal{R}$, stationary sound sources are constantly emitting audio waveform. We use two receivers to record the audio at various discrete positions independently. At each time step, we temporally synchronize the two receivers before recording so that the two receivers are recording the same audio content. In addition to audio, we also record the two receivers' spatial position. Specifically, we have collected $N$-step paired receiver dataset $\{(\mathcal{A}, \mathcal{P}) = \{(x_{1,i}(t), p_{1,i}), (x_{2,i}(t), p_{2,i})\}_{i=1}^{N}\}, \mathcal{A} \in \mathbb{R}^{T \times N}, \mathcal{P} \in \mathbb{R}^{3 \times N}$. The recorded audio $x_{1,i}(t)$ and $x_{2,i}(t)$ is the raw audio waveform in time domain (both are of the same length $T$ sampled at the same sampling rate). $p_{1,i}$ and $p_{2,i}$ are the two receivers' spatial coordinate $[x'_{k,i}, y'_{k,i}, z'_{k,i}]$ ($k \in \{1, 2\}$). Our target is to learn a receiver-to-receiver acoustic neural warping field $\mathcal{F}$ from two receivers position and audio $(\mathcal{A}, \mathcal{P})$, $\mathcal{F} \leftarrow (\mathcal{A}, \mathcal{P})$, so that it can efficiently predict the spatial audio acoustic effects for any arbitrary target position $p_t$ by predicting a warping transform $\mathcal{W}_{p_r \to p_t}$ that warps audio recorded at another reference position $p_r$ to the target position $p_t$. The warped audio $\widehat{x_{p_r \to p_t}}(t)$ at position $p_t$ essentially accommodates the spatial acoustic effects belonging to $p_t$.

$$\mathcal{W}_{p_r \to p_t} = \mathcal{F}_\theta(p_t, p_r); \quad \widehat{X_{p_r \to p_t}}(f) = \mathcal{W}_{p_r \to p_t} \cdot X_{p_r}(f); \quad p_t, p_r \notin \mathcal{P} \tag{1}$$

Where $\theta$ is the trainable parameters of $\mathcal{F}$. $p_r, p_t$ are arbitrary positions in the 3D space $\mathcal{R}$. $\widehat{X_{p_r \to p_t}}(f)$ and $X_{p_r}(f)$ are discrete Fourier transform (DFT) representation of the warped audio at the target position and recorded audio $x_{p_r}(t)$ at reference position represented in time domain, respectively. For example, if $x_{p_r}(t)$ has $T$ points, the $T$-point DFT result $X_{p_r}(f)$ is a complex representation where both the real and imaginary part have $T$ data points. The learned $\mathcal{W}_{p_r \to p_t}$ is a complex representation

of the same shape of $X_{p_r}(f)$. It is worth noting that, in our formulation, the warping transform is multiplication in frequency domain (see Sec. 3.2 for the proof), and the acoustic neural warping field is independent on audio content $\mathcal{A}$ (see Sec. 3.3 for more discussion).

To optimize $\mathcal{F}_\theta$, we minimize the discrepancy $\mathcal{L}$ between the warped audio at the target position and corresponding recorded audio,

$$\mathcal{F}_\theta \leftarrow \arg\min_\theta \; \mathcal{L}(\widehat{X_{p_1 \to p_2}}(f), X_{p_2}(f)), \quad \forall p_1, p_2 \in \mathcal{P} \tag{2}$$

## 3.2 MATHEMATICAL BACKEND OF RECEIVER-TO-RECEIVER NEURAL WARPING FIELD

Before introducing *SPEAR*, we need to answer two questions:

**Q1:**. *Does the proposed warping field suffice to model receiver-to-receiver spatial acoustic effects?*
**Q2:**. *If so, is there any constraint on the audio source number and placement in the 3D space?*

We first consider the simplest case where there is just one audio source in the 3D space.

***Proposition 1:*** If the Linear Time-Invariant (LTI) 3D space contains a single audio source, then receiver-to-receiver warping exists and is uniquely defined for any pair of receiver positions.

***Proof:*** Assume the audio source isotropically emits sound waveform $s(t)$ at a fixed position, the two receivers' recorded spatial audio in time domain at two different positions are $x_1(t)$ and $x_2(t)$, respectively. According to room acoustics (Savioja & Svensson, 2015) and if we assume the room is linear time invariant (LTI), $x_1(t)$ and $x_2(t)$ are obtained by convolving with their respective impulse response RIR with the sound source $s(t)$,

$$x_1(t) = s(t) \circledast h_1(t), \quad x_2(t) = s(t) \circledast h_2(t) \tag{3}$$

where $h_1(t)$ and $h_2(t)$ are the two RIRs from the source $s(t)$ to receiver $x_1(t)$ and $x_2(t)$ respectively. $\circledast$ is the 1D convolution in time domain. According the Convolution theorem that time domain convolution equals to production in Frequency domain, we can rewrite Eqn. (3) as,

$$X_1(f) = S(f) \cdot H_1(f), \quad X_2(f) = S(f) \cdot H_2(f) \tag{4}$$

where $X(\cdot)$, $S(\cdot)$ and $H(\cdot)$ are the Fourier transform of receiver recorded audio, source audio and RIR, respectively. Based on Eqn. (4), we can further get,

$$X_2(f) = X_1(f) \cdot \frac{H_2(f)}{H_1(f)}, \quad X_1(f) = X_2(f) \cdot \frac{H_1(f)}{H_2(f)} \tag{5}$$

Let $\mathcal{W}_{1 \to 2} = \frac{H_2(f)}{H_1(f)}$ (or $\mathcal{W}_{2 \to 1} = \frac{H_1(f)}{H_2(f)}$), we can conclude that: 1) the receiver-to-receiver warping universally exists and 2) is uniquely defined for any pair of receiver positions, 3) its existence just relies on the 3D space and is independent on if the audio source is emitting sound or not.

***Proposition 2***: If the more than one audio sources are placed in the 3D space, the receiver-to-receiver warping field existence is not guaranteed.

***Proof:*** Assume $K$ ($K > 1$) audio sources are placed in the 3D space, based on the superposition property in room acoustics (Savioja & Svensson, 2015), one receiver recorded audio (*e.g.*, $x_1(t)$) can be expressed as,

$$x_1(t) = s_1(t) \circledast h_{1,1}(t) + s_2(t) \circledast h_{2,1}(t) + \cdots + s_K(t) \circledast h_{K,1}(t) \tag{6}$$

where $h_{k,l}(t)$ indicates the RIR from the $k$-th source to the $l$-th receiver, By extending Eqn. (6) to $M$ receivers and further applying Fourier transform, we can get,

$$\begin{bmatrix} X_1(f) \\ X_2(f) \\ \cdots \\ X_M(f) \end{bmatrix} = \begin{bmatrix} H_{1,1}(f) & H_{1,2}(f) & \cdots & H_{1,K}(f) \\ H_{2,1}(f) & H_{2,2}(f) & \cdots & H_{2,K}(f) \\ \cdots & \cdots & \cdots & \cdots \\ H_{M,1}(f) & H_{M,2}(f) & \cdots & H_{M,K}(f) \end{bmatrix} \cdot \begin{bmatrix} S_1(f) \\ S_2(f) \\ \cdots \\ S_K(f) \end{bmatrix} \tag{7}$$

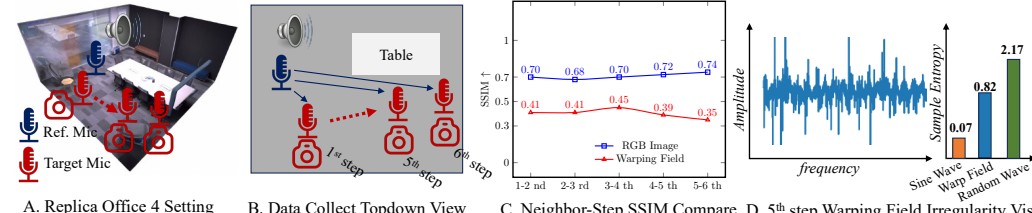

A. Replica Office 4 Setting    B. Data Collect Topdown View    C. Neighbor-Step SSIM Compare    D. 5th step Warping Field Irregularity Vis.

Figure 2: Two challenges in *SPEAR* learning: **Position-Sensitivity** and **Irregularity**. The position-sensitivity is represented by much lower structural similarity index (SSIM) of two neighboring-step warping fields than the two RGB images (sub-fig. C). The warping field irregularity is represented by both warping field visualization in frequency domain (real part) and much higher sample entropy score than regular sine wave (and just half of random waveform) (sub-fig. D).

For brevity, we can rewrite Eqn. (7) as $X = H \cdot S$. Since we have no knowledge of audio sources $S$, we can treat Eqn. (7) as multivariate polynomial linear function for audio sources $S$. The identifiability of $H$ can be potentially analyzed by independent component analysis (ICA). In certain conditions, $H$ is identifiable and full-rank, then the warping field exists for $K$ sources To ensure a unique solution for $S$, the determinant of the coefficient matrix $H$ must be non-zero ($\det(H) \neq 0$), and the rank of coefficient matrix $H$ must be equal to the rank of the augmented matrix $[H|X]$, rank$(H)$ = rank$([H|X])$. Moreover, even if we can deterministically represent audio source by receivers, $S = H^{-1} \cdot X$, we can hardly predict one receiver's spatial acoustic effects by warping from another single receiver because one receiver's spatial acoustic effects, in multiple audio sources case, usually depend on multiple other receivers. We empirically show one such example in Appendix A.

### 3.3 LTI RECEIVER-TO-RECEIVER WARPING FIELD PHYSICAL PRINCIPLE

We present three room acoustics physical principles (Kuttruff, 1979; Rayleigh & Lindsay, 1945) that will guide *SPEAR* design.

*Principle 1,* **Globality:** Unlike a normal RGB image just captures a localized area, a receiver recorded spatial audio relates to the whole 3D space. Originating from the source position, the spatial audio propagates in a complex way that incorporates reflection, diffraction and absorption before reaching to the receiver position, resulting in the interaction with almost the whole 3D space before reaching to the receiver. Consequently, the final receiver recorded audio is influenced by the whole 3D space.

*Principle 2,* **Order Awareness**: The Order Awareness principle states that *SPEAR* should account for the specific order of the input two receivers. In essence, the learned warping field varies when the order of the two receivers is swapped. This can be readily proved by Eqn. (5), since $\mathcal{W}_{1 \to 2} = \frac{H_2(f)}{H_1(f)}$, $\mathcal{W}_{2 \to 1} = \frac{H_1(f)}{H_2(f)}$, $\mathcal{W}_{1 \to 2} \neq \mathcal{W}_{2 \to 1}$.

*Principle 3,* **Audio-Content Agnostic**: This principle asserts that the receiver-to-receiver warping field is an inherent characteristic of the 3D space, affected by neither the presence or absence of audio within the 3D space nor the specific class of audio.

### 3.4 POSITION-SENSITIVITY AND IRREGULARITY OF WARPING FIELD

The complex behavior of spatial audio propagation in an enclosed 3D space often results in different spatial acoustic effects even for audio recorded at neighboring positions. This position-sensitivity becomes even more pronounced in the receiver-to-receiver warping field, where even a small receiver position can lead to substantial warping field variation. As is shown in Fig. 2, we compare the visual differences and warping field variations caused by small receiver position change. Using the 3D space `Office 4` from the Replica dataset (Straub et al., 2019) and the SoundSpaces 2.0 simulator (Chen et al., 2022), we place a stationary audio source and a reference receiver. Next, we ask the robot carrying a pinhole camera and a receiver to walk straightforward with step size 0.3 m in the vicinity of the reference receiver. At each target receiver position (crimson color in Fig. 2), we capture RGB images and compute the corresponding warping fields. We then adopt structural similarity index (SSIM) (Wang et al., 2004b) to measure the visual and warping field differences between two neighboring positions. The results, shown in Fig. 2, clearly indicate that a 0.3-meter position change

results in a more pronounced warping field variation (much lower SSIM score) compared to the RGB images. To show warping field irregularity, we visualize the 5-th step warping field real part in frequency domain in Fig. 2 D, from which we can see the warping field is highly irregular and thus exhibits higher sample entropy score (Guzzetti et al., 2000)[1] than regular sine wave.

## 3.5 SPEAR NEURAL NETWORK INTRODUCTION

The way we design *SPEAR* neural network is guided by Sec. 3.3 and Sec. 3.4. Specifically, *SPEAR* takes as input two receivers' positions (one reference position and the other target position) and outputs the corresponding warping field that warps the spatial acoustic effects at the reference position to the target position (**Audio-Content Agnostic** principle applies). We build *SPEAR* on top of Transformer architecture (Vaswani et al., 2017) and predict the warping field in frequency domain, in which the warping field is jointly represented by a real and imaginary series. Predicting the warping field in frequency domain results in both faster processing and better generation quality, as shown in the ablation experiment 4.7. To accommodate the **Globality** principle and tackle the position-sensitivity challenge, we construct a learnable grid feature spanning to the whole 3D space horizontally, each cell of the grid thus corresponds to one particular physical position in the 3D space. The two input positions' features are extracted from the grid feature by bilinear interpolation. Concatenating the two interpolated features in order (**Order-Awareness** principle satisfied) gets the

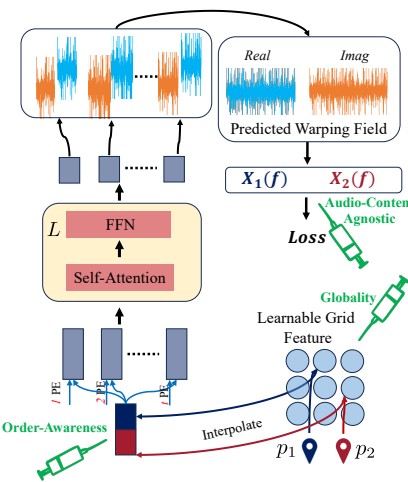

Figure 3: *SPEAR* network visualization.

input position-pair's representation. Each token then combines the input positions' representation and token index position encoding as the initial token representation. After 12 transformer blocks, the final learned token representation is further fed a prediction head to predict their corresponding real/imaginary warping field. We visualize the neural architecture in Fig. 3. The detailed layer parameters and feature sizes are given in the appendix D.

**Train and Inference.** To ensure the learned a general and universal warping field that can handle arbitrary audio in real scenarios, we adopt sine-sweep audio (He et al., 2024; Gao et al., 2020) covering the whole frequency range [0-16] kHz during the training phase . Since just one audio source presents in the 3D space, we can obtain the ground truth warping field by dividing the target position received audio by source position received audio (see Eqn. (5)). We thus jointly train *SPEAR* with group truth warping field and two receivers' recorded audio. The original warping field length we predict is 32 k. Due to the warping field symmetry (DFT conjugate symmetry), we just predict half warping field which are 16 k points. For ground truth warping field supervision, we combine both L1 and L2 loss. For the two receivers recorded audio supervision, we adopt spectral convergence loss (Arık et al., 2019). During training, we calculate the sum of L1 and L2 loss for both the real and imaginary part following (Zou & Hastie, 2005; Howard et al., 2021). Let ground truth and predicted warping field's real and imaginary part be $W_{gt}^{real}$, $W_{gt}^{imag}$, $W_{pred}^{real}$, and $W_{pred}^{imag}$, the combined L1 and L2 loss is calculated as $L(W_{pred}^{real} - W_{gt}^{real}) + L(W_{pred}^{imag} - W_{gt}^{imag})$, where $L(a, b) = \|a - b\|_1 + \|a - b\|_2^2$.

## 4 EXPERIMENT

### 4.1 DATASETS

We evaluate *SPEAR* on three datasets across different domains.

1. **Synthetic Dataset.** We adopt Pyroomacoustics (Scheibler et al., 2018) to simulate a large shoebox-like 3D space of size $[5m \times 3m \times 4m]$. All the receivers are placed on the same height, the audio source is placed at position $[2m, 2m, 2m]$. We simulate 3000 receivers positions for training and

---

[1]The higher of sample entropy score, the more irregular is the series. To highlight warping field irregularity, we compare its sample entropy score with totally regular sine wave and random wave.

another 300 receivers for test, by ensuring the reference-target receiver pair position in the test set is significantly different from those in training set.

2. **Photo-realistic Data.** To show our model is capable of predicting the warp field in a more complicated and photo-realistic environment, we employ Replica `Office 0` and `Office 4` 3D space (Straub et al., 2019) to simulate the spatial audio. In `Office 0` and `Office 4`, we simulate 4000 receivers and 8000 receivers for training, respectively. Both scenes have 500 receiver positions for test.

3. **Real-World Dataset.** We further test on the real-world MeshRIR (Koyama et al., 2021) dataset. We select a grid of $[21 \times 21]$ receivers from the MeshRIR-S32 dataset split to construct our dataset. 397 of the audio positions are used for training, and the rest 44 positions are reserved for test.

## 4.2 DATA PREPARATION

**Ground Truth Warping Field Acquirement** We generate the ground truth receiver-to-receiver warping field by dividing the target receiver audio by the reference receiver audio in frequency domain. However, this division operator inevitably leads to `NaN` value or abnormally large value ($> 100$, when the denominator is close to zero) in the obtained warping field (see the ground truth warping field visualization in Fig. V), resulting in the difficulty of accurately warping field learning. To address this issue, we make two adjustments. Firstly, we replace `NaN` values with zeros so that the whole neural network is trainable without encountering `NaN`, which could significantly harm the learning. Secondly, we clip all warping field values to lie within $[-10, 10]$. Our core intuition for adopting the `clip` operation is that the abnormally large values (outlier) easily allure the whole neural network to be trapped in predicting those abnormally large values, thus ignoring the learning of the warping field with normal values (inliers). We empirically verify this in Fig. V that `clip` operation is able to remove the outliers in the warped target audio.

**Data Sampling Strategy** For all three datasets, we construct train and test datasets by first splitting the receiver positions into two disjoint sets. The reference and target receiver positions are sampled from the same receiver position set. Intuitively, this ensures that the model never sees a warping field that warps audio from a receiver position in the training set to a position in the test set for fair evaluation. More detailed descriptions of sampling strategies for each dataset are given as follows.

For the synthetic data generation, we arrange the receivers in an $80 \times 40$ grid, with adjacent receivers spaced 0.05 meters apart. To create the test set, we select test samples in such a way that no two samples are adjacent in the grid. This interleaved sampling strategy ensures that each test receiver position is at least 0.05 meters away from any receiver position in the training set. For the photo-realistic data generation, due to the existence of furniture and other objects as obstacles in the room scenes, we do not employ the grid-sampling strategy used in synthetic data generation. Instead, we randomly sample 4500 and 8500 receiver positions on the two scenes' floors and select a subset of 500 receiver positions from each scene as the test set. The test set sampling strategy for Real-World data is the same as the interleaved sampling strategy used in the synthetic data generation.

## 4.3 EVALUATION METRICS

To quantitatively evaluate the warping field, we adopt: 1. **MSE**, mean square error between ground truth and *SPEAR* predicted warping field in frequency domain (average between real and imaginary part). 2. **SDR** (signal-to-distortion ratio). Following (Richard et al., 2022), we report SDR to assess the fidelity of predicted warping field. 3. **PSNR** (Peak Signal-to-Noise Ratio (Wang et al., 2004a)) and 4. **SSIM** (structural similarity index measure (Wang et al., 2004b)) to quantify the quality of the predicted warping field. We also introduce human-perceptual metric to provide insight into human perceptual quality of the predicted warping field. Specifically, we choose five speeches from VCTK (Yamagishi et al., 2019) dataset and warp them to the target position with the learned warping field, then compute the 5. **PESQ** (perceptual evaluation of speech quality (Wang et al., 2022)) score for the warped speech and ground truth recorded speech.

## 4.4 COMPARING METHODS

Given the novel problem setting of *SPEAR*, currently there is no existing method that directly applies to our problem. Existing RIR-based spatial acoustic effects modelling methods (Ratnarajah et al., 2021b;a; 2023) vary significantly in the way they exploit RIR data and the amount of prior room scene acoustic properties knowledge to train their model, we thus find it difficult to modify them to

Table 1: Quantitative result on three datasets. MSE (↓), SDR (↑), PSNR (↑), SSIM (↑), PESQ (↑).

| Method | Synthetic Data | | | | | Photo-Realistic Data | | | | | Real-World Data | | | | |
|---|---|---|---|---|---|---|---|---|---|---|---|---|---|---|---|
| | SDR | MSE | PSNR | SSIM | PESQ | SDR | MSE | PSNR | SSIM | PESQ | SDR | MSE | PSNR | SSIM | PESQ |
| LinInterp | −0.92 | 1.57 | 14.08 | 0.85 | 1.38 | −0.94 | 1.44 | **14.71** | 0.63 | 2.16 | −0.54 | 1.45 | 14.91 | 0.88 | 1.65 |
| NNeigh | −4.19 | 3.36 | 14.13 | 0.83 | 1.29 | −2.87 | 2.22 | 12.13 | 0.64 | 1.89 | −3.64 | 3.09 | 15.03 | 0.87 | 1.63 |
| NAF | 0.42 | 1.16 | 14.24 | 0.90 | 1.51 | 0.07 | 1.13 | 14.21 | 0.73 | 1.92 | 1.20 | 0.96 | 15.27 | 0.93 | 2.05 |
| *SPEAR* | **0.87** | **1.04** | **14.87** | **0.91** | **1.53** | **0.66** | **1.03** | 14.57 | **0.75** | **2.18** | **1.38** | **0.94** | **15.75** | **0.93** | **2.35** |

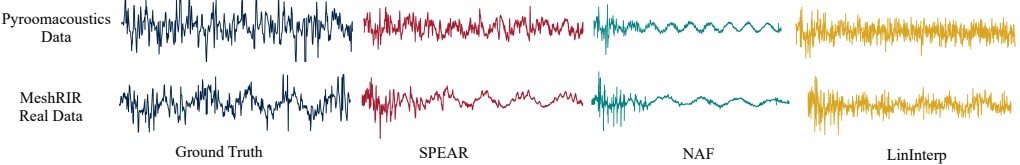

Figure 4: Learned warping field visualization on synthesized dataset (top) and MeshRIR real-world dataset (bottom). We just visualize the warping field real part. Complete visualization is given in Fig. II in Appendix.

fit our setting. For meaningful comparison, we compare *SPEAR* with one source-to-receiver neural acoustic field learning method (NAF (Luo et al., 2022)) and two other interpolation methods.

1. **NAF** (Luo et al., 2022). NAF learns continuous source-to-receiver RIR field by assuming access to massive RIR data. We modify it to accept two positions as input and output the warping field in frequency domain.

2. **LinInterp**: Neighboring Linear-Interpolation. For each receiver-pair in test set, we retrieve top-25 warping field data from the training set whose position are closest to the receiver pair (the position distance is defined by the the sum of reference position shift and target position shift). Then we average the 25 warping field to get the linear-interpolated warping field.

3. **NNeigh**: Nearest Neighbor method searches for the warp field, where the reference and target receiver positions are closest to the input test position pair. This is equivalent to retrieving the top-1 closest warping field from the training set.

### 4.5 IMPLEMENTATION DETAIL

We adopt AdamW optimizer (Loshchilov & Hutter, 2019) for training on all datasets. On the synthetic dataset, the model requires approximately 6000 epochs to converge, which takes around 7 hours on a single A10 Nividia GPU. We set the learning rate of the learnable grid feature to 1e-5, and the rest learnable parameters' learning rates to 1e-4. Using a smaller learning rate for the grid features improves the model training stability. Since the model predictions rely solely on the grid feature extracted, changes in grid feature can result in significant differences in model

| Method | Inf. Time | Param. Num. |
|---|---|---|
| NAF (2022) | 0.13 s | 1.61 M |
| *SPEAR* | 0.0182 s | 27.26 M |

Table 2: Model parameter and inference time comparison. The inference time is the average of 1000 independent inferences with batch size 32 on a single A10 GPU.

prediction. Therefore, setting a lower learning rate for the grid features prevents the model prediction from changing abruptly, and thus improves training stability.

Though our model has larger parameter size, the inference time is smaller than the NAF baseline. As shown in Table 2, our model has more than ten times larger parameter size than the NAF model, but inference speed is around ten times faster than the NAF model. This advantage stems from our transformer's architecture, which allows our model to generate the warping field in a single forward pass. In contrast, the NAF model predicts the warping field value at each frequency separately. This means 16 k forward passes are required to predict one full warping field, resulting in substantially slower prediction.

### 4.6 EXPERIMENTAL RESULT

The quantitative result on the three datasets is given in Table 1, from which we can see that *SPEAR* outperforms all the three comparing methods by a large margin. Among all the five metrics, *SPEAR* maintains as the best-performing method (except for one PSNR metric). Moreover, *SPEAR* outperforms NAF (Luo et al., 2022) significantly on metrics like SDR and PESQ.

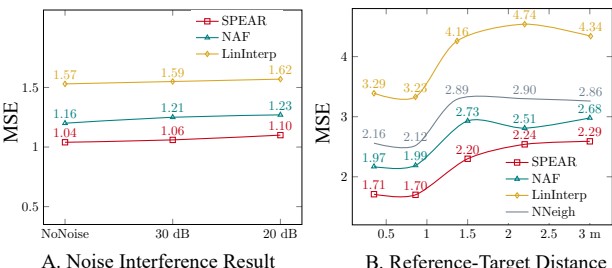

A. Noise Interference Result

B. Reference-Target Distance

Figure 5: Ablation Study on noise interference (A), reference-target receiver distance (B).

| Metrics | Chirp | Engine | Person | Siren |
|---------|--------|--------|--------|--------|
| PSNR | 14.866 | 14.867 | 14.799 | 14.902 |
| SDR | 0.869 | 0.869 | 0.866 | 0.868 |
| SSIM | 0.907 | 0.907 | 0.897 | 0.898 |
| MSE | 1.044 | 1.045 | 1.043 | 1.043 |

Table 3: Ablation study result on *SPEAR* audio-content agnostic characteristic.

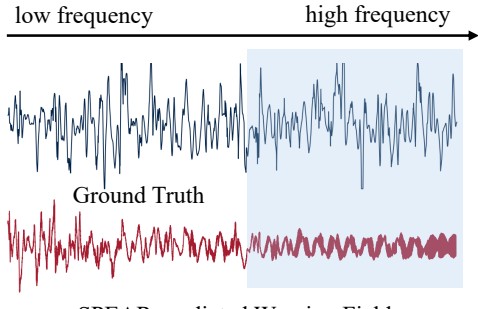

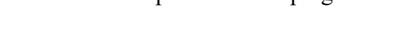

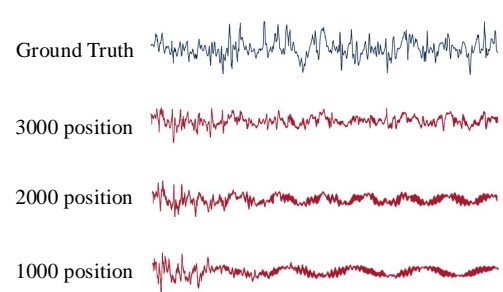

Figure 6: *SPEAR* predicted failure warping field on one Pyroomacoustics synthesized data. The area in light blue indicates the high-frequency region.

Figure 7: Ablation Study result on sampled position number. We observe that using reduced sampled position number to train *SPEAR* leads to reduced warping field prediction accuracy.

We provide qualitative comparison of the predicted warping field in Fig. 4, in which we visualize the warping field (real-part) on Synthetic dataset (top row) and MeshRIR real-world dataset (bottom row) obtained by all methods. In all warping field visualization plots, the warping field values are shown in increasing frequency order from left to right. From this figure, we can clearly observe that 1) while the ground truth warping field exhibits high irregularity (see Sec. 3.4), *SPEAR* is capable of learning the irregularity pattern from the input position pair; 2) The two other methods (NAF (Luo et al., 2022) and LinInterp) failed to tackle the irregularity challenge. NAF (Luo et al., 2022) tends to predict close warping field values. The higher of the frequency, the easier it tends to predict the same value, the left and right part comparison of NAF). We further provide visualization of predicted warping field imaginary parts in Fig. II in Appendix.

We further illustrate a failure case in Fig. 6, where *SPEAR* was applied to a Pyroomacoustics-synthesized dataset. In this figure, it is evident that *SPEAR* struggles to accurately predict the warping field in the high-frequency region (highlighted in blue). Additionally, we observe that *SPEAR* tends to degenerate, predicting values within a narrow range. Unlike the incapability in high-frequency part, *SPEAR* can predict more accurate warping field in the low frequency area. This phenomena is more pronounced for NAF (Luo et al., 2022) (see Fig. 4). It further remains as a future research direction to investigate the challenge of predicting accurate warping field in high frequency range. More failing cases are shown in Fig. III and Fig. IV in Appendix.

### 4.7 ABLATIONS

We run all ablation studies on Synthetic dataset created by Pyroomacoustics (Scheibler et al., 2018).

**Warping Field Sensitivity to Noise.** We add two ambient Gaussian noises (measured by SNR, 20 dB and 30 dB) to test model's performance under noise interference. As is shown in Fig. 5 A, we can observe that while all three comparing methods have seen performance drop (increased MSE metric) as more noise is involved, *SPEAR* maintains as the best-performing method and outperforms the other two methods (NAF and LinInterp) by a large margin under all noise interference.

**Warping Field Accuracy w.r.t. Reference-Target Receiver Distance.** We further test *SPEAR*'s capability in predicting spatial acoustic effects for spatially distance target receivers. To this end, we compute models' performance variation w.r.t. reference-target receivers' distance change ($0.5\ m -$ $3.0\ m$) and show the result in Fig. 5 B. We can observe from this figure that: 1) The increased reference-target receiver's distance leads to performance drop (increasing MSE) for all methods, which is expected as the spatial acoustic effects drastically change when the receiver position gets changed dramatically. 2) *SPEAR* still outperforms the other comparing methods, showing its advantage in predicting warping field for further target receivers.

**Warping Field Prediction in Frequency V.S. Time domain.** In *SPEAR*, we propose to predict warping field in frequency domain for computation efficiency concern. Out of the computation efficiency concern, we further want to figure out the performance with predicting in time domain. As depicted in Fig. 8, we can observe that predicting warping field in time domain leads to significant performance drop as the model tends to predict all-zero warping field.

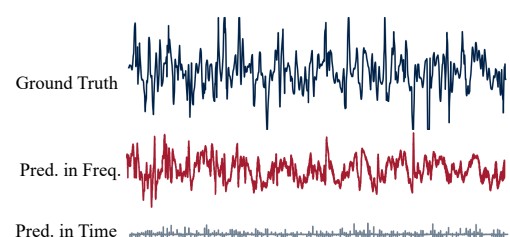

Figure 8: Ablation Study result on prediction on frequency versus time domain.

**Audio-Content Agnostic Verification.** Since all models are trained with Sine Sweep audio, we want to know if the *Audio-Content Agnostic* principle truly satisfies. To this end, we evaluate *SPEAR* with another three audios: engine, siren and human vocalization. As is shown in Table 3, we can verify that *SPEAR* is agnostic to audio content and can be applied to arbitrary audio class.

**Effect of Sampled Position Size on Model Training Performance** In all three datasets, reference and target positions are densely sampled. In this section, we verify the necessity of dense sampling in order to learn reasonably good warping field. To this end, we randomly sub-sample 1000, 2000 positions from the whole 3000 sampled positions for training that we used in the main experiment. With the subsampled positions, we re-train the *SPEAR* model and further evaluate the model on the same test data. The quantitative result in Table 4 shows that the model

| # Sample Position | SDR | MSE | PSNR | SSIM |
|---|---|---|---|---|
| 3000 | 0.87 | 1.04 | 14.87 | 0.91 |
| 2000 | 0.49 | 1.14 | 14.33 | 0.90 |
| 1000 | 0.36 | 1.17 | 14.11 | 0.89 |

Table 4: The impact of different sampling density on model performance. MSE ($\downarrow$), SDR ($\uparrow$), PSNR ($\uparrow$), SSIM ($\uparrow$), PESQ ($\uparrow$)

performance drops significantly as the sampling size reduces. We further visualize one comparison of the warping field predicted by *SPEAR* model trained with smaller sampled data size in Fig. 7, more visualizations are given in Fig. VI in Appendix. From these figures, we can clearly obverse that reduced sampled position size inevitably leads to worse warping field prediction. This large sampled position size requirement shows the core challenge of the receiver-to-receiver acoustic neural warping field learning because the warping field is position sensitive (as we presented in Sec. 3.4). We leave it as a future research direction to explore novel learning methodology with much less position size.

## 5 CONCLUSION AND LIMITATION DISCUSSIONS

We introduce a novel receiver-to-receiver spatial acoustic effects prediction framework that can be trained in a data much more readily accessible way. We theoretically prove the existence and universality of such warping field if there is only one audio source. When there are multiple sources distributed in the room scene, the warping field may not exist and we can apply source separation method (Petermann et al., 2023) first to separate the sources. Moreover, once the audio source position gets changed, the whole *SPEAR* needs to be retrained. There are two limitations that remain to be resolved in the future. The first is that we still require dense sampling to get reasonably good performance. Second, we assume all receivers' position lies on the same horizontal plane. More in-depth investigation is needed to remove this constraint and allow receiver placement in the whole 3D space.

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

## A  APPENDIX

## A  RECEIVER-TO-RECEIVER WARPING FIELD EXISTENCE DISCUSSION

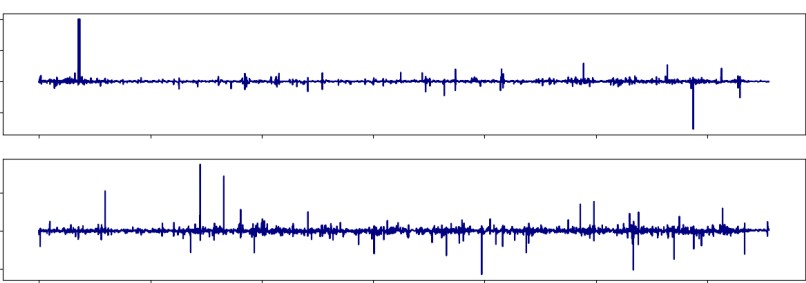

Figure I: Warping field visualization in frequency domain (real part). Top: warping field with two audio sources `Engine` and `Footstep` two audio sources. Bottom: warping field with the two audio sources at the same positions but the `Footstep` sources is replaced by `Telephone Ring`.

In this section, we empirically show that the receiver-to-receiver neural warping field may not exist if more than one audio sources present in the 3D space. To this end, we depend on Pyroomacoustics Scheibler et al. (2018) simulator to simulate a shoe-box like 3D space of shape $[5 \times 5 \times 5]$ meters. Two audio sources are placed at coordinate A $[1, 1, 1]$ meters and B $[4, 4, 4]$ meters, respectively. Two receivers are accordingly placed at coordinate $[1, 1, 2]$ meters and $[4, 4, 3]$ meters respectively. In the fist simulation, we place `Engine` audio at source position A and `Footstep` audio at source position B. In the second simulation, we just replace the `Footstep` audio at position B with `Telephone Ring`. With the two receivers recorded audio, we depend on Eqn. (5) to calculate the warping field from the reference receiver at $[1, 1, 2]$ to the target receiver at $[4, 4, 3]$. The computed two warping fields are shown Fig. I, we clearly see that the two warping fields are significantly different. We thus can conclude that the receiver-to-receiver neural warping field is no longer solely dependent on receiver positions (so the **Audio-Content Agnostic** principle does not apply). When more than audio sources present in the 3D space, the warping field proposed in this work may not exist.

## B  DISCUSSION ON ACOUSTIC NEURAL WARPING FIELD VISUALIZATION

The acoustic neural field is represented in frequency domain, each of which in our case contains a real and imaginary one-dimensional data vector. In the main paper, we just visualize the real part due to the space limit. Here, we provide another five real/imaginary warping fields visualization in Fig. II. From this figure, we can clearly see that our proposed framework *SPEAR* is capable of handling the warping field irregularity property.

## C  FAILURE CASE VISUALIZATION

During the experiment process, we find all methods inevitably give failure case warping field predictions. We visualize part of some failure cases predicted by *SPEAR* on both synthetic data (Fig. III), photo-realistic and real-world dataset (Fig. IV). As we discussed in the main paper, the position-sensitivity and irregularity pose challenges in the warping field prediction. We hope these failure cases will attract more investigation into this research problem.

## D  *SPEAR* NETWORK ARCHITECTURE

*SPEAR* network architecture is shown in Tab. I.

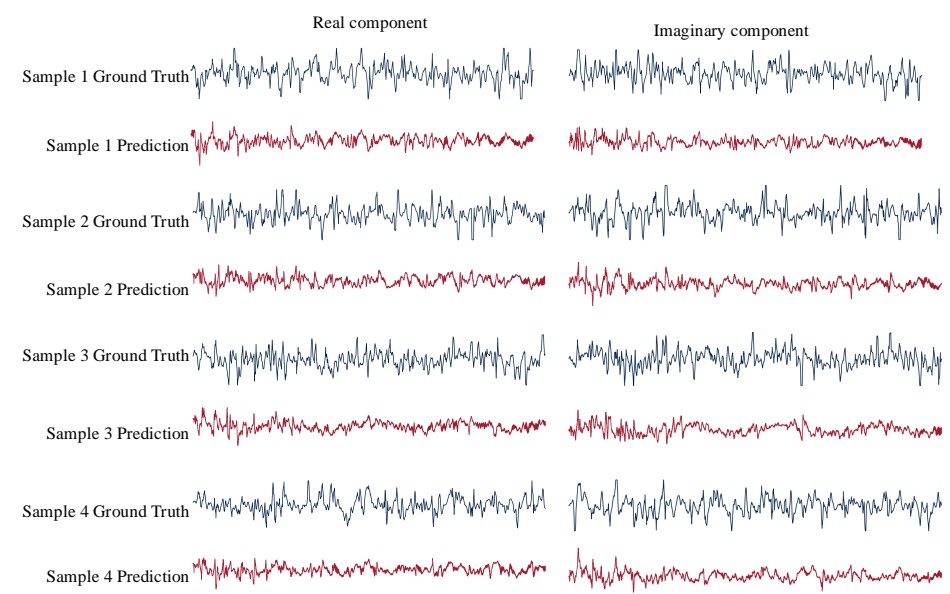

Figure II: Visualization of real and imaginary component of the ground truth warp field in the synthetic dataset.

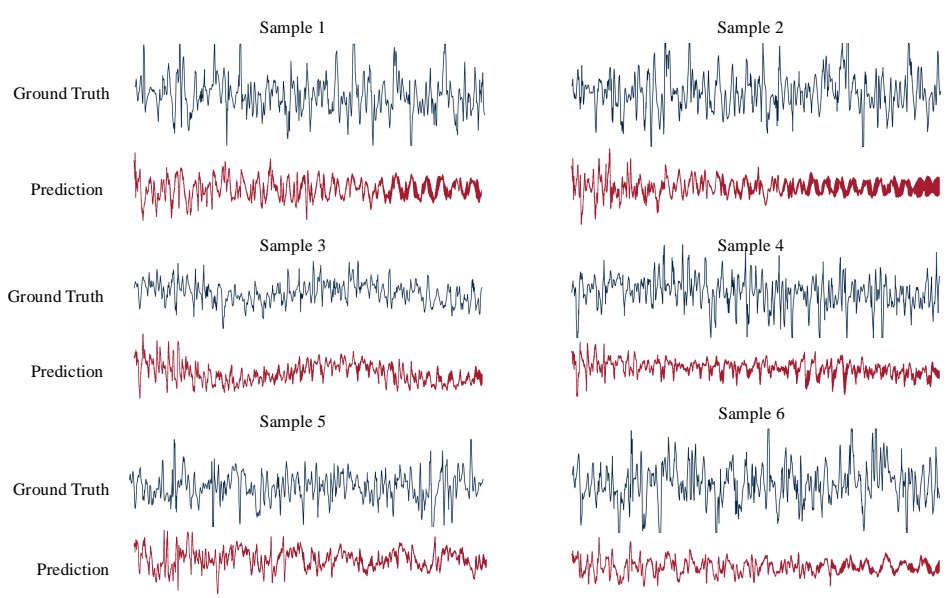

Figure III: Examples of failure cases of SPEAR model on synthetic data.

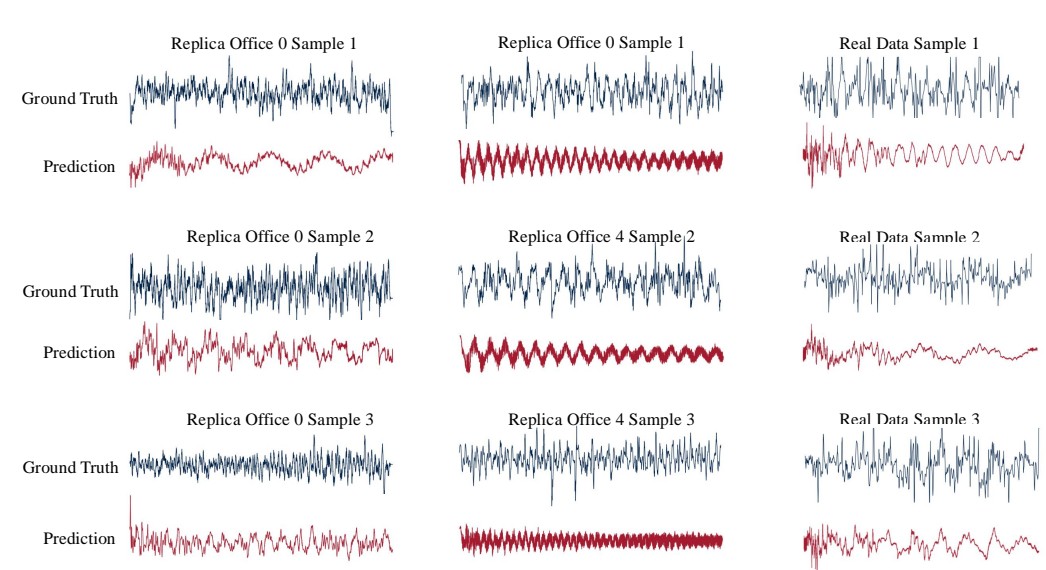

Figure IV: Failure case visualization of *SPEAR* model on both Photo-realistic and Real-world Dataset.

| Layer Name | Filter Num | Output Size |
|---|---|---|
| **Model Input**: 2 position 3d coordinate: [2, 3] | | |
| **Grid Feature**: concatenated 2 position feature: [1, 384] | | |
| **Transformer Encoder Input**: Initial Token Representation: [43, 384] | | |
| Transformer Layer 1 | head num = 8, hidden dim = 384 | [43, 384] |
| ... | ... | ... |
| Transformer Layer 12 | head num = 8, hidden dim = 384 | [43, 384] |
| **Prediction Head** | | |
| Real part FC | FC, output_feat = 384 | [43, 384] |
| Imaginary part FC | FC, output_feat = 384 | [43, 384] |
| Flattern | Flattern real/imaginary token sequence. Construct complex sequence. | [16512] |
| Prune | Cut the sequence to 16384 length | [16384] |
| Mirroring | Generate the full warping field by concatenating the predicted sequence with its mirrored conjugate sequence | [32768] |

Table I: *SPEAR* Network Architecture Detail.

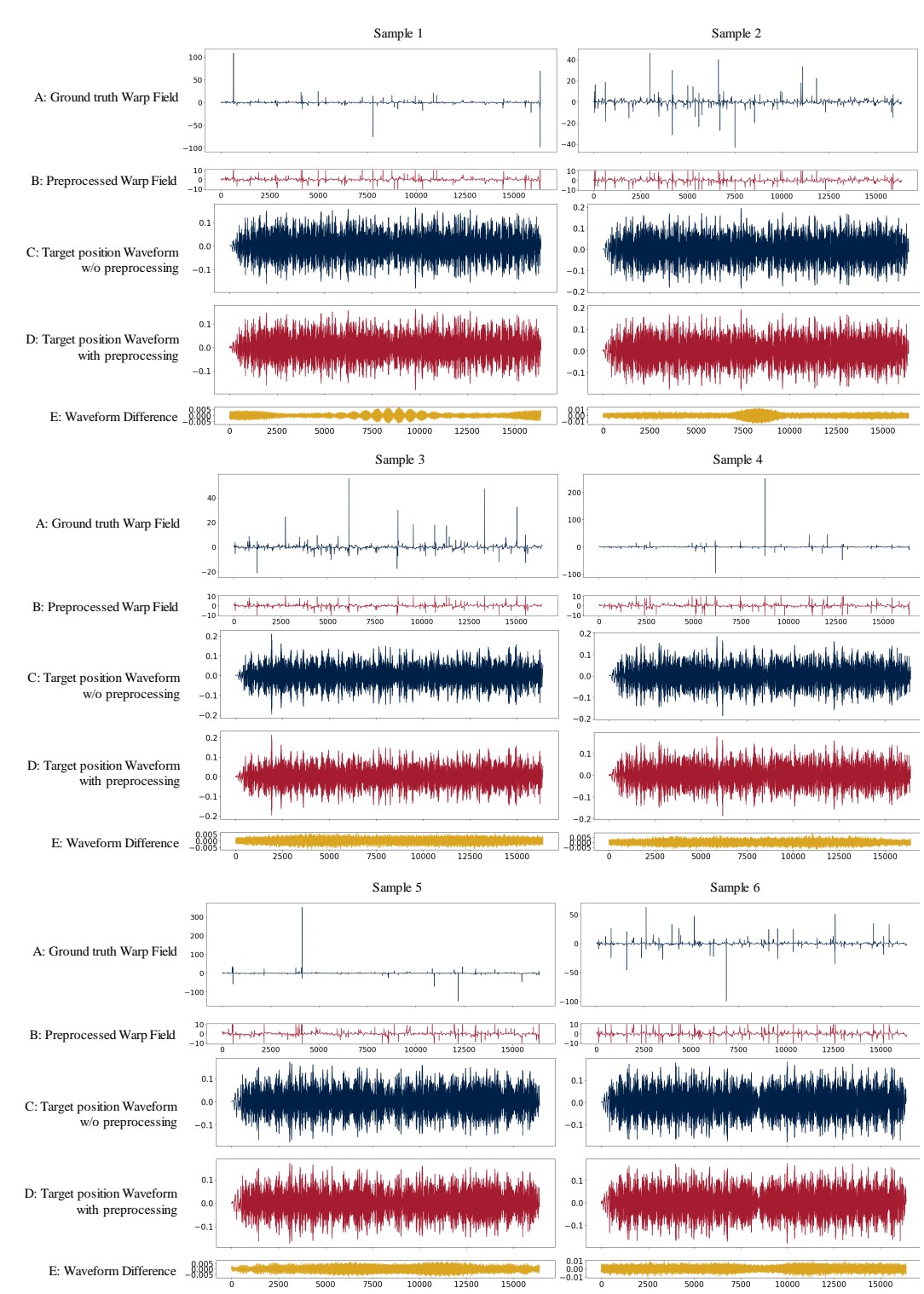

Figure V: Visualization of the effect of preprocessing Warping field on the generated audio waveform. In plot A C, we show the warping field and waveform of the target position without warping-field preprocessing. In plot B D, we show the pre-processed warping field and the waveform at the target position after applying the preprocessed warping field. Plot E shows the difference between the two waveforms at the target position.

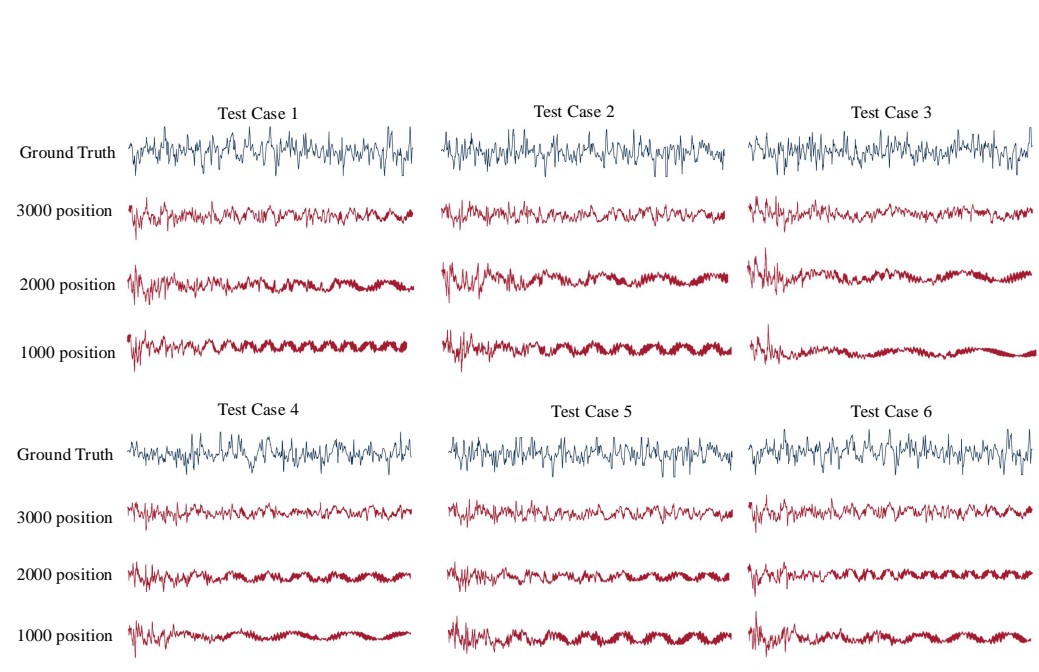

Figure VI: Predictions of models trained with 3000/2000/1000 sample positions in the training dataset.

