# OpenReview forum: "SPEAR: Receiver-to-Receiver Acoustic Neural Warping Field"
_ICLR.cc/2025/Conference — Submitted to ICLR 2025_

### Official Review · Reviewer_RxZ5 · 2024-10-30

**Soundness:** 2
**Presentation:** 2
**Contribution:** 1
**Rating:** 3
**Confidence:** 4

**Summary:**

The authors point out that the problem of acoustic field estimation is characterized by the high cost of collecting prior information (about the physical properties such as geometries), and propose a relatively simple way to solve the problem: to estimate the receiver signal at another location from the receiver signal. They present their theoretical assumptions and backgrounds for their methodology and demonstrate its effectiveness on simulated and real recorded datasets.

**Strengths:**

One of the strengths of this paper is its value as an attempt to alleviate the difficulty of data acquisition for sound field synthesis. It also deserves recognition for achieving large computational efficiency gains within moderate memory efficiency.

**Weaknesses:**

For me, the justification for the main contribution of this study remains unclear. To point out my thoughts on the main contributions listed by the authors at the end of Section 1:

1. Please clarify the reasoning behind the claim that the receiver-to-receiver framework requires less prior information than source-to-receiver (e.g. NAF).

    - At the end of the second paragraph of Section 1, the authors explain their motivation as collecting RIR data is exceedingly difficult in real scenarios, and their receiver-recorded audio is more readily accessible. What I don't understand about this argument is that they ended up using a sine sweep source for training, so in what way is this more efficient than training with RIR?

    - Let's say that the training data is not necessarily a sine sweep. One of the cases where a receiver-to-receiver framework can be beneficial, as the paper suggests, is when we don't know about the source signal. I'm curious to hear your thoughts on whether cases like this are prevalent in real-world scenarios. In my understanding, as there should be a receiver recording to train SPEAR in a certain room, one should first play the source signal to acquire the responses at each receiver point, meaning that we already know the source signal.

2. The statements and proofs of the propositions in Section 3.2 seem to lack the rigor to be presented as major contributions.

    - What the authors describe as the claim and proof in the text for Proposition 1 is a direct consequence of Rayleigh's reciprocity theorem. Aside from the rigors in the proof of the “unique existence of warping given any pair of receivers”, there is something I don't understand about what the authors conclude in the last sentence of the proof: what does the claim “independent of the audio source” mean? Considering diffraction, the warping field should be variant depending on the source frequencies, isn’t it?

    - Similarly, there are several technical misleading statements throughout the paper that could be used to claim theoretical contributions, and the explanation of acoustic propagation is still underdeveloped. One example of this is the claim in Principle 1 in Section 3.3 that 'the entire 3D space is involved in the receiver recording', which could be misleading to those unfamiliar with audio. It would be better to claim instead that 'because the sound is generally more reflective than light, even signals reflected from occluded areas can be affected, and because of their lower velocity, the effect is more pronounced'.

3. To claim to have revealed the superiority of SPEAR, the baseline selection is weak, and the ablation study did not sufficiently reveal the strengths and weaknesses of the proposed methodology.

    - At the very least, INRAS [1] should be included, and it would be nice to see other baselines such as AV-NeRF [2], DiffRIR [3], etc.

    - In the context of synthesizing acoustic warping fields for moving sources with a fixed receiver (which shares systematic similarities with this paper’s problem statement), WaveNet-based architectures are often used [4,5], or even MLPs [6] to estimate the frequency domain warping fields. How does the Transformer architecture bring advantages over these other architectures?

    - I wonder if re-training is essential for comparing performance with RIR-based source-to-receiver methodologies (including NAF). Even with keeping those source-to-receiver models as-is, we can estimate $H_{p_r}$ and $H_{p_t}$ directly, and it seems natural to be able to obtain $\mathcal W_{p_r \to p_t}=H_{p_t}/H_{p_r}$. Is there a reason to retrain the NAF nonetheless? Given such commutative nature of LTI systems, how is receiver-to-receiver learning systematically different from source-to-receiver learning, as one could readily get the impulse response for each of the receivers and then deconvolve one into the other?

[1] Su, K., Chen, M., & Shlizerman, E. (2022). Inras: Implicit neural representation for audio scenes. Advances in Neural Information Processing Systems, 35, 8144-8158.
[2] Liang, S., Huang, C., Tian, Y., Kumar, A., & Xu, C. (2023). Av-nerf: Learning neural fields for real-world audio-visual scene synthesis. Advances in Neural Information Processing Systems, 36, 37472-37490.
[3] Wang, M. L., Sawata, R., Clarke, S., Gao, R., Wu, S., & Wu, J. (2024). Hearing Anything Anywhere. In Proceedings of the IEEE/CVF Conference on Computer Vision and Pattern Recognition (pp. 11790-11799).
[4] Richard, A., Markovic, D., Gebru, I. D., Krenn, S., Butler, G. A., Torre, F., & Sheikh, Y. (2021). Neural synthesis of binaural speech from mono audio. In International Conference on Learning Representations.
[5] Leng, Y., Chen, Z., Guo, J., Liu, H., Chen, J., Tan, X., ... & Liu, T. Y. (2022). Binauralgrad: A two-stage conditional diffusion probabilistic model for binaural audio synthesis. Advances in Neural Information Processing Systems, 35, 23689-23700.
[6] Lee, J. W., & Lee, K. (2023, June). Neural fourier shift for binaural speech rendering. In ICASSP 2023-2023 IEEE International Conference on Acoustics, Speech and Signal Processing (ICASSP) (pp. 1-5). IEEE.

**Questions:**

A few minor comments based on the weaknesses mentioned above.

- For the claim “to be more favorable to real-world scenarios” to be convincing, the method would need to be validated in cases where the source signal is not a sine sweep, but a more readily accessible sound like clapping (which is easy to generate and record at the receivers but difficult to acquire the original source). Furthermore, the performance under different receiver position sampling policies would need to be more rigorously validated to see how much space is covered by the receiver positions for training. In this respect, the choice of dataset in this paper could benefit from further diversification:

    - Pyroomacoustics was synthesized based on the image-source method in a shoebox room as stated in the paper

    - Sound Spaces 2.0 is also based on ray-based simulation and the office subset of Replica used by the authors is a single-room

    - MeshRIR is a real measurement, but the structure of the room is also a shoebox

    A dataset that can verify performance for non-trivial cases may be a multi-room dataset (where there could be no direct acoustic path, e.g., 'line of sight'), such as Replica's Apartment subset. In this case, other factors than the Reference-Target Distance reported in this paper may be important for sampling receiver positions. For example, if the source is in the living room of an apartment, how much detail should the training receiver be sampled to ensure performance?

- Typos/misleading phrases
    - L182, L205: The part where the author expresses "according to room acoustics (Savioja & Svensson, 2015)" seems to assume an LTI system, but since not all room acoustics assume LTI, the expressions seem to need clarification.
    - L370 PSESQ $\to$ PESQ
    - Figure 8 comes before Figure 6 and 7.
    - Wrong references to "Fig. B" (L456) and "Fig. C" (L464)

---

> ### Author Response · Authors · 2024-11-24
> **Feedback 1 to reviewer #RxZ5**
>
> We sincerely thank you for your review and suggestions on improvements.
>
> **Weaknesses part**
>
> **Q1**: Please clarify the reasoning behind the claim that the receiver-to-receiver framework requires less prior information than source-to-receiver (e.g. NAF).
>
> **A1**: Thank you for raising this concern. The argument that "receiver-to-receiver framework requires less prior information than source-to-receiver (e.g. NAF)" is supported by two key points:
>
> 1. SPEAR is source-agnostic, it doesn't need to know the source position (which is required by source-to-receiver methods). This property is especially helpful when we enter into a new environment and we have no knowledge about the source position. Our observation, getting to know the precise sound source position sometimes is difficult task (it refers to another sound source detection and localization task). With SPEAR, we can directly "go-and-collect" data collection method is convenient to deploy.
>
> 2. SPEAR doesn't require RIR data. RIR data collection is very tedious and low-efficient. It requires the 3D space to be silent, and requires two humans (or two robots) to physically move to different positions to collect RIR by sending and receiving spatial sound (like gun-shot, balloon explosion).
>
> **Sine-Sweep Sound Training**, we involve Sine-Sweep training so as to cover the sound whole frequency, since our goal is train a general and universal receiver-to-receiver warping field that can be directly adopted to different sound, and we verified this in Table 3 and Sec. 4.7. It is easy to understand sound with either low-frequency or high-frequency to train, the model can just handle the fed frequencies and becomes incapable of handling other frequencies. It is worth noting that adopting since-sweep sound is widely adopted in "acoustic active sensing"[1][2].
>
> [1] Y. He, et al., Deep Neural Room Acoustics Primitive, ICML-24.
>
> [2] R. Gao, et al., VisualEchoes: Spatial Image Representation Learning through Echolocation. ECCV-20.
>
> **Sine-Sweep Prevalence Concern**, we agree that sine-sweep is not always available in real scenario. So during training phase, we require the sound source to emit "special" sound. We argue that this is an unavoidable step for both receiver-to-receiver method and source-to-receiver method [2] as long as to learn a general and universal neural field (not a collapsed field tailored for special sound). In source-to-receiver methods, the required RIR data collection also requires to send special sound (gun-shot, sine-sweep or balloon explosion) in order to fully capture the RIR. Moreover, we just require Sine-sweep audio during training phase. In testing, SPEAR can handle arbitrary sound, so it can process arbitrary prevalent sound.
>
> **Q2**: Proposition 1 is a direct consequence of Rayleigh's reciprocity theorem.  What does the claim “independent of the audio source” mean?
>
> **A2**: As far as we understand, Proposition 1 isn't a direction consequence of Raleigh's reciprocity theorem. First, Raleigh's reciprocity theorem deals with source-to-receiver's RIR. It states that the resulting RIR from position A to position B is the same as the RIR from position B to position A. However, proposition 1 shows the unique existence of warping field if and only if one source exists. The warping field from A to B is different from that swapped warping field from B to A. We don't rely on Raleigh's reciprocity theorem to derive proposition 1.
>
> In the statement, "independent of the audio source" indicates the receiver-to-receiver warping field existence is independent on if the sound source is emitting sound. Even if the 3D room scene is silent, the warping field still exists. We have refined the expression in the revised paper.
>
> **Q3**: several technical misleading statements throughout the paper.
>
> **A3**: We really appreciate you pointing this out. In our revised version, we have carefully revised those misleading statements to make it clear for readers from non-acoustics background.
>
>
> **Q4**: To claim to have revealed the superiority of SPEAR, the baseline selection is weak, and the ablation study did not sufficiently reveal the strengths and weaknesses of the proposed methodology.
>
> **A4**: Thanks for pointing this out. We are following your advice to add more baseline comparisons, we will report the result once they are ready.

---

> ### Author Response · Authors · 2024-11-24
> **Feedback 2 to reviewer #RxZ5**
>
> **Q5**: How does the Transformer architecture bring advantages over these other architectures, like MLP and WaveNet.
>
> **A5**: Thanks for asking the motivation behind the Transformer selection. We have partially answered similar question in reviewer \#zARm **Q9/A9**. Here we would like thoroughly discuss it.
>
> In fact, we have tried a number of network architecture before deciding to use the Transformer architecture. For example, we have tried,
>
> 1. NeRF [3] motivated MLP (multi-layer perceptron) to encode the warping field, in which the MLP neural network takes two positions (after either position encoding or full-connection to embed position to high dimension) as input and outputs a feature. The learned feature is added with warping field frequency index position encoding to predict the warping field one-by-one. It turns out predict all-zero warping field.
>
> 2. EMD (empirical mode decomposition) based warping field prediction. The motivation is decompose the highly irregular warping field into a set of simple and smooth warping fields, and predict each relatively smooth warping field separately. Again, we found the neural network (also MLP based) can just predict over-smooth warping field, and still fail at complex warping field.
>
> 3. MLP based RIR prediction framework, including Fast-RIR [4] and and TS-RIR [5]. All of them tend to predict all-zero warping field.
>
> 4. To tackle the irregularity of the warping field, we also proposed to predict each value by combining raw localization (via classification, which initially localize the value within a raw range) and fine-grained small offset prediction (via regress head). It turned out to predict all-zero warping field.
>
> 5. We also tried several latest time-series prediction neural network, and found they couldn't handle the irregularity property of the warping field.
>
> 6. We also considered WaveNet but finally abandoned it because WaveNet is designed for audio generation, the input and output are both audio waveform. But in our case, the input are two receiver positions. We have tried to feed the two receivers' positions to position encoding to get high-dimensional representation and treat it as input "pseudo-waveform" to WaveNet. It turned out that WaveNet also learn all-zero warping field.
>
> We gradually finalize to adopt Transformer architecture after lots of prior attempts. Our assumption on why Transformer can predict reasonably good warping field includes: 1. using tokens to represent the warping field allow to implicitly capture the complex long-range dependency along the frequency axis, which is naturally achieved by multi-head self-attention layer. 2. asking each single token to predict a localized and non-overlapping warping field enables to neural network to predict a localized warping field in a relatively independent manner, and they were not confused or affected by global predictions.
>
> [3] Ben Mildenhall et al., NeRF: Representing Scenes as Neural Radiance Fields for View Synthesis, ECCV 2020.
>
> [4] Ratnarajah, A et al., Fast-RIR: Fast Neural Diffuse Room Impulse Response Generator, ICASSP, 2022.
>
> [5] Ratnarajah, A et al., TS-RIR: Trans-lated Synthetic Room Impulse Responses for Speech Augmentation. ASRU, 2021.
>
> **Q6**: If re-training is essential for comparing performance with RIR-based source-to-receiver methodologies.
>
> **A6**: Thanks for pointing this out. Theoretically, we can get the receiver-to-receiver warping field by first getting the source-to-receiver RIR. But it goes against the assumption that the sound source is agnostic and we can't get the RIRs. This is why we don't directly compare with RIR-based models by directly asking those model to predict RIRs, but instead choose to re-train them. However, you have pointed out an important follow-up research direction that we have also considered about: Since dividing two RIRs operation inevitably leads to increased warping field irregularity difficulty, we can avoid the division operation by instead predicting two quasi-RIR for each receiver respectively. The quasi-RIR is not real RIR because it is also source-agnostic. We are currently exploring this research direction and hopefully it will mitigate the difficulty in learning the warping field.

---

> ### Author Response · Authors · 2024-11-24
> **Feedback 3 to reviewer #RxZ5**
>
> **Questions part**
>
> **Q7**: Method would need to be validated in cases where the source signal is not a sine sweep, but a more readily accessible sound like clapping. Furthermore, the performance under different receiver position sampling policies would need to be more rigorously validated to see how much space is covered by the receiver positions for training.
>
> **A7**: Thanks for mentioning this point. The reason why we use sine-sweep source signal during training phase is illustrated in **Q1**. We would like to emphasise that adopting sine-sweep sound to train a full-frequency general receiver-to-receiver warping field that is capable of handling arbitrary sound during testing phase that just cover partial frequency. We can train SPEAR with non- sine-sweep sound, but it will collapse into the specific sound carried frequency. We discussed and show the generalization of sine-sweep sound trained SPEAR in Table 3 and Sec. 4.7.
>
> **Sampling Policy Discussion**: For Replica dataset, we used random sampling strategy due to the fact the indoor furniture will make some sample points (like above furniture points) can't be sampled. For MeshRIR and Pyroomacoustics based datasets, we adopted grid-like sampling strategy. Currently, we have to insightful observations: 1. Receiver-to-receiver warping field is irregular even in flat area, a small position change will be leading to obvious warping field change. Form RIR perspective, this is partially due to the position sensitivity of RIR (sound propagation complex), the fraction of the RIRs inevitably generates extra irregularities. 2. The warping field becomes more irregular in areas close to the wall or complex layout. In other words, these areas require higher dense samplings. We agree that more in-depth investigation on sampling policy is needed, we can't finish all of them due to the rebuttal time limit. In addition to searching for better sampling policy, we have also provided several potential new methods that deserve attention. (See our feedback to Reviewer \#y6xA, Q5/A5).
>
> **Q8**: Typos and misleading phrases and clarifications.
>
> **A8**: Thanks for letting us know. We have carefully corrected them and added clarifications in the revised version.

---

> ### Author Response · Authors · 2024-11-25
> **Feedback 4 to reviewer #RxZ5**
>
> We modified the INRAS model's input and evaluated its performance on the Synthetic Dataset. In particular, we place 50 surface points on the perimeter of the scene in equal intervals, which results in a $0.32m$ spacing between adjacent surface points. Since the sound source location is not available during training, we replaced the scatter module with a learnable feature of shape $50 \times 256$, where $256$ is the dimension of feature associated with each of the $50$ surface point. The rest of the model architecture, including layer number, neuron numbers in each layer, and activation function, remain the same as the original INRAS model. The comparison between INRAS's result and other methods reported in the main paper is shown below:
>
> | Method | SDR  | MSE  | PSNR  | SSIM | PESQ |
> |--------|------|------|-------|------|------|
> | LinInterp | -0.92 | 1.57 | 14.08  | 0.85 | 1.38 |
> | NNeigh | -4.19 | 3.36 | 14.13  | 0.83 | 1.29 |
> | NAF | 0.42 | 1.16 | 14.24  | 0.90 | 1.51 |
> | SPEAR | 0.87 | 1.04 | 14.87 | 0.91 | 1.53 |
> | INRAS  | 0.01 | 1.28 | 14.18 | 0.89 | 1.46 |
>
> INRAS architecture outperforms non-learnable baseline methods in all evaluation metrics. However, the model fails to achieve better performance than the NAF and the proposed SPEAR model. In addition, we tested the model parameter size and the inference time. The inference time is the average of 1000 independent inferences with batch size 32 on a single A10 GPU, measured in the same way as that used in the main paper.
>
> | Method | Param. Num | Inf. Time |
> |------|------|------|
> | NAF | 0.13 s | 1.61 M |
> | SPEAR | 0.0182 s | 27.26 M |
> | INRAS | 0.0702 s | 0.8 M|
>
> In comparison to NAF, which also exploited MLP as the main model backbone, INRAS achieves both smaller parameter size and higher inference speed. This advantage makes it favourable to be deployed in portable devices that require fast inference.

---

> ### Comment · Reviewer_RxZ5 · 2024-11-28
> **Official Comment by Reviewer RxZ5**
>
> Thank you for your time and efforts to address the concerns I raised. While I agree with some of your answers and I found many improvements were made in the revisions, I still have some remaining concerns.
>
> **Receiver-to-receiver framework validity**
>
> 1. Specifically which part of the paper shows that “SPEAR is source agnostic”? It sounds like SPEAR should also work well for any source conditions, but it contradicts the fact that SPEAR does not generalize for all source positions. Actually, it was hard for me to find that SPEAR is only trained/tested for a fixed source position, and the only part that states (from the revised version) is a single line: "once the audio source position gets changed, the whole SPEAR needs to be retrained". I would greatly appreciate seeing the investigations about 'how the performance drops' rather than making possibly misleading statements to highlight performance.
>     - The **fact that the source position was fixed for all datasets** should be stated in Section 4.1 as the source position sampling is indeed considered as a critical factor in the receiver-to-receiver framework.
>     - If the source position was fixed per each dataset, (again) what is the advantage of having the receiver-to-receiver model over the source-to-receiver model as we need a new SPEAR model for every source position?
>         - Keeping in mind the fact that source position is fixed, I wonder (again), what makes it unfair to (directly) compare SPEAR with other source-to-receiver models (without re-training)?
>     - I think having to retrain SPEAR for the change of source positions is a natural consequence, as it is claimed to be content-agnostic where the change in the acoustic field could be captured from the content. What I would like to know includes the following examples:
>         - Let’s just assume the shoe-box room, we fix the receiver pairs and then run SPEAR by changing the source positions. I believe the warping field should change as both the source-to-receiver paths change. Would the performance stay the same regardless of where the source is? How does SPEAR know such change in the acoustic field despite being both source-agnostic and content-agnostic?
>         - Let’s assume that we have a coupled room (say, room A and room B) and trained SPEAR with training data where the source is only sampled from room A. Then how would the test results vary depending on whether the source is in room A or room B  (and also depending on the receiver pair sampling policy)?
>
>
> 2. How sine-sweep acquisition is different from RIR acquisition, and what exactly are the advantages? To me, they seem almost identical other than using either the sweep or the impulse.
>     - What the authors mention in the **Feedback to All Reviewers** about SPEAR that it “just requires two robots (or humans) holding a microphone to record the acoustic environment” is not true, as it always should accompany a sine sweep which is not quite different from a usual RIR measurement other than that SPEAR requires two microphones.
>
> **Mathematical Rigors in Proofs**
>
> Can you please expand specifically on which part of the proof mathematically shows the ‘uniqueness’ and the ‘existence’ of the “warping field” in Proposition 1? For example, can you elaborate on: In which space the “warping field” is defined? Which line assures its existence? How do you show that it is unique?
>
> Just to briefly mention why I think it is a direct consequence of the reciprocity theorem: I believe the logic follows the propagation $\text{receiver}_1 \to \text{receiver}_2$ is built upon the concatenation of two LTI systems as $\text{receiver}_1 \to \text{source} \to \text{receiver}_2$ which can be directly inferred from the reciprocity theorem (as you are using $\text{receiver}_1 \to \text{source}$ from the fact that you know $\text{source} \to \text{receiver}_1$) which is not quite surprising to know. I would appreciate it if the authors could provide more mathematical rigors in their proof, to claim that “theoretically proving” is the main contribution of this paper.

---

> ### Author Response · Authors · 2024-11-29
> **Further Feedback 1 to Reviewer RxZ5**
>
> We thank the reviewer for the further comments and remaining concerns. We feel priviledged to continue to receive your comments.
>
> ### **Receiver-to-Receiver Framework Validity**
>
> 1. **Concern:** SPEAR is only trained/tested for a fixed source position. **Feedback:** To understand this, we can think a 3D space where we have a receiver position A and position B. For the source position $S_1$ and another source position $S_2$, the resulting warping field $W_{A\rightarrow B, S_1}$ from pos A to pos B for $S_1$ should be different from the resulting warping field $W_{A\rightarrow B, S_2}$ from pos A to pos B for $S_2$, $W_{A\rightarrow B, S_1} \neq W_{A\rightarrow B, S_2}$. (it is quite straightforward because once the source changes, the RIRs to pos A and pos B gets changed, the warping field obtained by two RIRs' division gets changed accordingly). Therefore, we can't train SPEAR that takes the same two receiver position as input, but outputs different field, but instead require to train SPEAR for each source position separately. This  is different from source-to-receiver methods (or RIR-based methods) that can handle different source positions because they explicitly model source position in their methods. SPEAR is receiver-to-receiver, having no information about source position. So one SPEAR model sticks to one source position. We also provide mathematical proof of the potential non-existence of the warping field when multiple sources co-exist simultaneously. In conclusion, such "one source" constraint results from the receiver-to-receiver warping field formulation.
>
> 2. **Concern:** Advantage over source-to-receiver models. **Feedback:** First, we want to emphasize that in our goal in this paper isn't to show the advantage of SPEAR over source-to-receiver models, but instead to show a different way (or feasibility) to model spatial acoustic effect for a given receiver position. There are pros and cons of SPEAR when comparing to RIR-based methods. **Pros**: SPEAR doesn't require the source position (although it requires source speaker emits sine-sweep audio during training phase, it still doesn't require source position info) and sometimes getting to know the source position is a difficult task; SPEAR can warp spatial acoustic effects from one receiver to another receiver (In fact, we were motivated by one scenario: two people using a phone to take a video by walking around a church independently, the church has church bell sound from one position. If one video happens to be audio-muted, luckily we can use SPEAR to warp the sound to the muted video from the unmuted video). **Cons**: As you raised, one SPEAR can just handle one-source.  Multiple sources requires multiple SPEAR models. It remains as a future research topic to investigate how to extend SPEAR to be naturally able to handle multiple sources. Our current idea is to decompose the warping field into sub warping fields.
>
> 3. **Concern:** would SPEAR performance stay as the same regardless of where the source is, or in room A and room B. **Feedback:** as we discussed above, SPEAR doesn't involve source info, so its performance just reflects the source upon which SPEAR is trained on.
>
> 4. **Concern:** How sine-sweep acquisition is different from RIR acquisition, and what exactly are the advantages? **Feedback:** You are right, RIR acquisition also requires the source sound to cover the whole frequency range and records the clean sound, so it requires source sound like sine-sweep and impulse and the 3D space to be noise-free (a quiet space). However, we want to highlight that the reason why we use sine-sweep sound is to incorporate a whole-frequency-range sound, we can adopt other sound as long as they cover the whole frequency range (for example, we have tested noise interference in Fig. 5 A, we can composite sine-sweep and other sound, like dog-barking, piano). Therefore, our data collection strategy and motivation is different from RIR acquisition, although we agree there are overlapping between them.
>
> ### **Mathematical Rigors in Proofs**
>
> 1. **Concern:** Expand specifically on which part of the proof mathematically shows the 'uniqueness’ and the ‘existence’ of the “warping field” in Proposition 1. **Feedback:** Thanks for pushing us to give more rigorous proof. We derive the proposition 1 mainly from LTI room acoustics (Eqn. 3, 4, 5): Once the source position, two receiver positions are fixed in a given 3D space, the RIRs from the source to the two receivers RIRs uniquely exists (Eqn. 3), the resulting warping field that is caculated by divising the two RIRs in frequency domain (Eqn. 4, 5) uniquely exists accordingly. In other words, the "uniqueness and existence" is derived from "uniqueness and existence" in LTI room aoucsitcs (RIRs).

---

> ### Author Response · Authors · 2024-11-29
> **Further Feedback 2 to Reviewer RxZ5**
>
> ### **Mathematical Rigors in Proofs**
>
> 2. **Concern:** reciprocity theorem concern.
>
> **Feedback:**
> * Thanks for further illustration of using reciprocity theorem to derive our proposition 1. In the main paper, we center around the source ($\text{receiver}_1 \leftarrow \text{source} \rightarrow \text{receiver}_2$) to mathematically prove proposition 1 (Eqn. 3, 4, 5). If we understand your idea correctly, you suggest to go through $\text{receiver}_1 \rightarrow \text{source} \rightarrow \text{receiver}_2$ pipeline. Based on reciprocity theorem, RIR($r_1\rightarrow s$) = RIR($s\rightarrow r_1$), so we can drive $\text{receiver}_1 \xrightarrow{\text{warpfield}}  \text{receiver}_2$ by $ \text{receiver}_1\xrightarrow{\text{RIR}(r_1\rightarrow s)} \text{source} \xrightarrow{\text{RIR}(s\rightarrow r_2)} \text{receiver}_2$. However, we find this chain rule (e.g. receiver1 via RIR1 to source, and via RIR2 to receiver2, or further via RIR3 to receiver3 (or source 3)), doesn't apply here because reciprocity theorem **1 just applies to a source-receiver pair, 2 requires to swap the source to the receiver (so the receiver then becomes source)**. Another intuitive way to understand this is that, the sound from $\text{receiver}_1$ to $\text{source}$ should be lounder than the sound at $\text{receiver}_1$. Applying RIR($r_1\rightarrow s$) can't achieve this. This is why we think directly applying reciprocity theorem can't rigorously get proposition 1 (because we didn't put the source at the $\text{receiver}_1$ position ).
>
>  * In our derivation, we implicitly followed the $\text{receiver}_1 \rightarrow \text{source} \rightarrow \text{receiver}_2$ pipeline to derive proposition 1, but we didn't depend on the reciprocity theorem. Instead, we choose to map the sound at $\text{receiver}_1$ to source by inversing the RIR($s\rightarrow r_1$) in frequency domain: $S(f) = \frac{X_1(f)}{H_1(f)}$ (where $H_1(f)$ is the RIR($s\rightarrow r_1$) representation in frequency domain) and further map the "source sound" to $\text{receiver}_2$ (see Eqn. 4, 5). It is worth noting that $\frac{1}{H_1(f)}$ isn't the RIR($r_1\rightarrow s$) derived from reciprocity theorem, and we temporally call it inverse-RIR. We also visualized the inverse-RIR both time time and frequency domain, it is much more complex and sophisticated than a normal RIR.
>
>  * Again, we apprecite you highlighting the reciprocity theorem. It is very intuitive and straightforward to describe the warping field through $\text{receiver}_1 \rightarrow \text{source} \rightarrow \text{receiver}_2$ chain. We also tried to decompose the warping field into two sub-warping fields: one **inverse warping field** mapping the $\text{receiver}_1$ sound to neural source space, then another **forward-warping field** to map from the neural sound space to $\text{receiver}_2$, but we finally didn't choose this workflow because we experimentally found the neural network hard to learn meaningful representation.
>
>    We hope we have correctly addressed your concern on reciprocity theorem. If you still have any concern, we are more than happy to discuss more.
>
> **Summary:** Unfortunately, we can't update any revised paper here. We are continuing to improve the paper writing and presentation by 1) explicitly providing more details on SPEAR training/testing process, data acquisition, requirements etcs. 2) revising Problem Formulation section to expliciting show SPEAR can only handle one-fixed source position, and doesn't require to know source position info. 3) clearly demonstrating the pros and cons between SPEAR and source-to-receiver based methods.
>
> Sould you have any other concern, you are sincerely invited to put it here and we are happy to answer more questions.

---

> ### Comment · Reviewer_RxZ5 · 2024-11-29
> **Official Comment by Reviewer RxZ5 on Further Feedback**
>
> Thank you for providing the feedback. I'm afraid I have to say that: I already understand the details that the authors explain above and still, my concerns have not been resolved. I am also aware that the authors are  To clarify my intention in the comment, my concerns are:
>
> 1. I strongly think the fact that **SPEAR is trained in a *fixed (specified position, stationary) source* should be clearly stated** (e.g. in Section 4.1) as I cannot agree that the “SPEAR model is source agnostic”. There are many factors in audio sources, such as location and directivity, but I think it is seriously misleading to claim that it is "source agnostic" without showing generalization across these factors.
>     - **Regarding Feedback 1**: What the authors describe in the “warping field formulation” (probably Proposition 1,2) is about one stationary source and the ill-posedness in multiple stationary sources and that was not my question. My concern is about SPEAR’s generalization in **one displaced source scenario** (so the scenario when the source is located at a different position), which the authors stated SPEAR does not generalize to in their revised manuscript, and my point is that the paper should emphasize enough about that if it is to claimed to be “source agnostic”.
> 2. I think the main contribution should be revised, if the authors are to keep claiming that "our goal in this paper isn't to show the advantage of SPEAR over source-to-receiver models", as it contradicts the main contribution stated at the end of the Introduction: “We propose SPEAR, a novel receiver-to-receiver spatial acoustic effects prediction framework. Unlike existing source-to-receiver modelling methods requiring extensive prior space acoustic properties knowledge, SPEAR can be efficiently trained in a data more readily accessible manner."
>     - **Regarding Feedback 2**: To me, it still seems like SPEAR simply assumed the source to be **fixed at a certain position** so that it does not need to care about it anymore. What I raised as a concern is not about multiple sources, my concern is that SPEAR cannot handle the change of source’s position but claims to be called as source agnostic.
> 3. I know how sine sweep covers the frequency ranges, and that is exactly why I am asking for further clarification. An impulse has all the energy of the entire frequency domain, condensed in a short time, and a sine sweep also has the energy of a wide frequency domain, but it usually changes the frequency slowly over time. My question is: The difference between RIR measurement (which you say is very inefficient) and sine sweep measurement (which you claim to have advantages) seems to be only whether the frequency-specific energy of the source signal is varied over time or simultaneously, so what are the rationales behind insisting on using a sine sweep as the source signal instead of an impulse, and what makes the RIR measurement inefficient compared to sine sweep?
>
>
> If the authors still believe that the "goal in this paper isn't to show the advantage of SPEAR over source-to-receiver models", then I cannot help but disagree with the main contribution 1, and even if we exclude the comparison between the two, I still have difficulty understanding what advantage the receiver-to-receiver scenario has. What is the big win of having a receiver-to-receiver model if one has to train a new SPEAR for a new source position and/or directivity even with sine-swept recordings as a training dataset?
>
> If the authors still believe that the mathematical contribution is the main contribution and at the same time that the proof of Proposition 1 is mathematically rigorous, then I cannot agree with that main contribution 2. Following the mention that the authors rely on the '"uniqueness and existence" in LTI room acoustics (RIRs)', my question is -- you did not show the uniqueness and the existence of 'RIR' $h$ used in equation (3) in your *proof*. How do you define your RIRs here after all? RIRs are **not** necessarily unique, and we can even make the statement 'RIRs do not exist' hold true depending on how we define their (mathematical) space/domain. I am concerned about this kind of mathematical rigor.

---

> > ### Author Response · Authors · 2024-12-02
> > **Further Feedback to the Official Comment by Reviewer RxZ5**
> >
> > Thank you for your further comments and concerns. We provide the following feedback here.
> >
> > 1. SPEAR is trained in a fixed (specified position, stationary) source should be clearly stated.
> >
> >    **Feedback:** Sure, we explicitly added this statement in our further revised version (however, we can't upload the reivsed version here). We will further mitigate the "source-agnostic" statement by specifying our requirements of the sound source other than its position (including the directivity you mentioned).
> >    * **concern is about SPEAR’s generalization in one displaced source scenario**: **Feedback:** We explicitly made this clear in the Introduction Section in the revised version -- we specify one trained SPEAR for a given 3D space each time just works for one source position. Once the source gets displaced, the SPEAR framework needs to be re-trained.
> >
> > 2. I think the main contribution should be revised.
> >
> >    **Feedback:** Thanks. We have revised it by adding more explanation that shows how we "define"the advantages here. In the original paper, we wrote: We propose SPEAR, a novel receiver-to-receiver spatial acoustic effects prediction framework. Unlike existing source-to-receiver modelling methods requiring extensive prior space acoustic properties knowledge, SPEAR can be efficiently trained in a data more readily accessible manner. Our initial idea is to show the advantage via the "data acuqisition" perspective. We have noted that traditional RIR estimation methods, both ray-based (treating sound as rays) and wave-based methods (depending on sound wave nature, and wave equations to derive the RIR) require to know lots of 3D space room acoustic properties (e.g., room geometric shape, constructional material, furniture layout, etc). Acquiring those acoustic properties sometimes is much challenging than SPEAR's data acquisition requirement (although we require the source to emit full-frequency-range sound during training phase, we specified this). Our original goal wasn't to show SPEAR's advantage over RIR-based method itself, but the amount of data acquisition effort.
> >
> >    **SPEAR cannot handle the change of source’s position but claims to be called as source agnostic.**: Thanks. We followed your suggestion and modified the "source-agnostic" statement.
> >
> > 3. What are the rationales behind insisting on using a sine sweep as the source signal instead of an impulse, and what makes the RIR measurement inefficient compared to sine sweep?
> >
> >    **Feedback**: As we just require the received sound in training data just cover the whole frequency range, both sine sweep and an impulse should work as long as they meet the "whole frequency range" requirement. The difference is that: in RIR measurement, we need to send-and-receive just- and clearn sine-sweep (or impulse) and further ensure the 3D space is silent. In our data collection process, we don't put such requirement and we even learn with noisy-sinesweep (for example, we have tested noise interference in Fig. 5 A) sound where noise or other normal sound (like piano) are added to the sine-sweep. That is to say, we think SPEAR data acquisition set more flexible constraint than RIR measurement.
> >
> > 4. mathematical contribution is the main contribution concern.
> >
> >     **Feedback:** we modified the mathematical contribution claim (especially regarding its rigorousness) in the current version. Based on your info, we are investigating the "uniqueness and existence" of RIRs in LTI room acoustics. Our understanding is that if the 3D space is LTI and static (no moving objects are moving around in the 3D space) and the sound source emits isotropically at a stationary position, the resultant RIR uniquely exists at any position that sound can traverse to. As to your mentioned "RIRs are not necessarily unique, and we can even make the statement 'RIRs do not exist' hold true depending on how we define their (mathematical) space/domain", we can investigating this and you are mostly welcome to give any related work so as to help us to continue to improve our work. We appreciate your strong support!

---

### Official Review · Reviewer_zARm · 2024-10-31

**Soundness:** 3
**Presentation:** 3
**Contribution:** 2
**Rating:** 3
**Confidence:** 4

**Summary:**

The paper presents a method for estimating the wrapping field in an enclosed acoustic environment, which is the relative transfer function between two receivers, when given the position of the two receivers. This method is meant to replace direct room impulse response estimation which relies on prior space acoustic properties and complex computations or requires massive amount of RIRs as a supervision. The method is shown to outperform baselines on both simulated and real-world data.

**Strengths:**

The paper presents a novel viewpoint that learns the wrapping field that connects two receivers using a new transfomer-based model. A comprehensive experimental study is conducted with both simulated and real-world data, and different aspects of the proposed method are examined. The paper is clearly written and contains meaningful illustrations that clarify its core ideas.

**Weaknesses:**

The usefulness of the proposed method is not well motivated. On the one hand, the wrapping field corresponds to a fixed source position, but in reality it is more informative to consider a source that can change locations. On the other hand, the space of all possible wrapping fields seems to be unnecessarily large, since it is defined by two receiver locations, whereas for RIR estimation the mapping is a function a single receiver only (and a source position). This is also evident from the vast amount of training samples that are required for training the model. Note that from these same measurements one can extract the RIR if the emitted source signal is known (which should be the case since the training recordings are performed in a controlled manner).
Note that this idea of utilizing the wrapping field already exists in the literature for two decades. It is known as the Relative Transfer Function (RTF). A plethora of methods have been proposed to robustly estimate the RTF from measured signals (see a summary on [1] Section IV. C. 3). More close to the current paper, it was already proposed in previous works to generate RTFs based on source-receiver locations [2,3,4]. This relevant literature should be referred to in the paper, and it should be made clear what is the difference between these works and the current contribution.
In addition, the paper is very similar to [5], where it seems that the main difference is that the current paper deals with RTF estimation instead of RIR estimation, thus requiring an additional emitting sound at a fixed position. It should be clarified what is the main contribution of the current work compared to [5].

[1]  Gannot, S., Vincent, E., Markovich-Golan, S., & Ozerov, A. (2017). A consolidated perspective on multimicrophone speech enhancement and source separation. IEEE/ACM Transactions on Audio, Speech, and Language Processing, 25(4), 692-730.

[2] Wang, Z., Vincent, E., & Yan, Y. (2017). Relative transfer function inverse regression from low dimensional manifold. arXiv preprint arXiv:1710.09091.‏

[3] Wang, Z., Li, J., Yan, Y., & Vincent, E. (2018, April). Semi-supervised learning with deep neural networks for relative transfer function inverse regression. In 2018 IEEE International Conference on Acoustics, Speech and Signal Processing (ICASSP) (pp. 191-195). IEEE.‏

[4] Bianco, M. J., Gannot, S., Fernandez-Grande, E., & Gerstoft, P. (2021). Semi-supervised source localization in reverberant environments with deep generative modeling. IEEE Access, 9, 84956-84970.‏

[5] He, Y., Cherian, A., Wichern, G., & Markham, A. (2024, January). Deep Neural Room Acoustics Primitive. In Forty-first International Conference on Machine Learning

**Questions:**

Several questions related to the weaknesses mentioned above:

1.	What is the motivation for learning the RTF where both receivers can change positions while source position is fixed (in test time maybe we would like the source to change places)?
2.	What is the relation to previous works on RTF estimation [2-4]?
3.	What is the key contribution over [5]?

Additional minor questions:

4.	Why is the Fourier transform used instead of STFT representation?
5.	Line 151 - the meaning of this notation $\mathcal{F}\leftarrow(\mathcal{A},\mathcal{P})$ is unclear.
6.	There is inconsistency in the notation, switching from $p_1,p_2$ to $p_r,p_t$ and the same for the wrapping field.
7.	Proposition 2 contains an over-general statement (“existence is not guaranteed”) and the proof is vague. In general, the identifiability of the mixing matrix in (7) was investigated under the field of independent component analysis (ICA), and there are certain conditions for which the matrix is identifiable (full rank matrix and at most one Gaussian source).  The wrapping field can be defined as a vector of length K that contains the wrapping field for each individual source.
8. Line 294: "The two input positions’ features are extracted from the grid feature by bilinear interpolation" - please clarify how the bilinear interpolation is performed.
9. What is the motivation for using a transformer? What does the attention between patches is expected to learn?
10. The proposed method is not necessarily more realistic compared to baselines. The required data is similar to collecting massive RIR data, since RIR can be extracted when the source signal is known.
11.	Fig 4., why NN baseline is not presented?
12.	The figures order does not follow the order they appear in the text -  Fig. 6 should come before Fig. 5.
13.	Warping Field Sensitivity to Noise - is the noise added during training?   What happens in higher noise levels?
14.	Missing details regarding experiments – what is the reverberation time? What is the signals length?
15.	Why Fig. 5.a. does not contain comparison to NN?
16.	Appendix D is empty
17.	Are there any insights regarding the characteristics of the failure cases?
18.	Table 3 – It is unclear why the feature dimension is 384, and what is 43 in the initial token representation. Why is there a pruning step at the output?

---

> ### Author Response · Authors · 2024-11-22
> **Feedback 1 to reviewer #zARm**
>
> Thank you for your constructive review and comments. We feel honoured to hear that you appreciate the comprehensive experimental study and clear writing of our work.
>
> **Q1**: Motivation for learning RTF with different receiver position and fixed sound source position.
>
> **A1**: Our model is effective in the following scenario: Suppose a few tourists uploaded their recorded audio of one scene, (e.g. audio taken in a concert or speech given in a chapel), and one other user wants to explore the scene online using the recorded audio. As this online user virtually moves in the scene, our model predicts the warping field that changes the acoustic effect from the ones recorded in known positions (uploaded by the tourists) to the audio heard at the virtual user's position. In these scenarios, the sound source position remains the same across audios recorded at different positions. Rendering audio effects using our end-to-end method does not need the precise coordinate of the sound source or the room impulse response recorded beforehand.
>
> **Q2**: Relation to previous works on RTF estimation.
>
> **A2**: Thank you for pointing this out. The prior works on RTF estimation have only attempted to predict the relative transfer function conditioned on the source position and the receiver pair position. Since the focus of these papers are on sound source localization, simple receiver configuration was used where the pair of receivers have fixed relative position and placed close to each other. This means their experiment result does not demonstrate the warping field prediction quality when receivers are separated by large distances. Our work allows arbitrary receiver pairs in the scene, which is more suitable for the application scenario described in Answer 1.
>
> **Q3**: Contribution over Deep Neural Room Acoustics Primitive (DeepNeRAP):
>
> **A3**: Thanks for mention this work. **Similarity**: Both SPEAR and the DeepNeRAP tend to model spatial acoustic effects that can be experienced at a spatial position by a receiver. Moreover, they both don't directly measure room acoustic properties such as constructional material and furniture layout, but instead actively probe the environment and implicitly characterize room acoustic properties. **Difference**: the fundamental difference between SPEAR and DeepNeRAP lies the way they choose to model room acoustic properties. While DeepNeRAP follows traditional room impulse response (RIR) based source-to-receiver pipeline, SPEAR adopts receiver-to-receiver pipeline. This difference leads to the different target (SPEAR learns a receiver-to-receiver warping field, DeepNeRAP learns source-to-receiver neural RIR), and different data collection methods (SPEAR requires two receivers to record sound, DeepNeRAP requires one source and one receiver separately). In summary, the contribution of SPEAR over DeepNeRAP lies in its novel receiver-to-receiver way to implicitly predict the spatial acoustic effects for an arbitrary receiver position, it is source-agnostic and warps the spatial sound from one position to the target position.
>
>
> **Q4**: Why use Fourier transform but not STFT.
>
> **A4**: Thank you for mentioning this. We have also experimented with predicting warping field for STFT instead of Fourier transforms of the audio. We didn't select this approach to ensure model's robustness to different sound source contents. If the sound emitted by the source contains a segment of silence, the audio recorded for the corresponding time frames will also be approximately silent. These time frames should be excluded from the warping field calculation, as it is meaningless to predict a warping field between two silent audio segments. Additionally, since the STFT frames associated with these silent segments are close to zero, calculating the warping field for these time frames will result in excessively large warping field values at all frequencies. Optimizing the model to predict such warping fields significantly affects its performance.
>
> **Q5**: Line 151 - the meaning of this notation $\mathbfcal{F}\leftarrow (\mathcal{A},\mathcal{P})$ is unclear.
>
> **A5**: Thanks for pointing this out. What we want to express in this notion is that the receiver-to-receiver acoustic neural warping field $\mathcal{F}$ is learned from collected two receivers' recorded sound $\mathcal{A}$ and positions $\mathcal{P}$. We add explanation in the refined version.
>
> **Q6**: There is inconsistency in the notation, switching from to $p_1, p_2$ to $p_r, p_t$ and the same for the wrapping field.
>
> **A6**: Thanks for mentioning this. We us $p_1, p_2$ to indicate two receivers, and $p_r, p_t$ to differentiate which receiver is the reference receiver $p_r$ and which receiver is the target receiver $p_t$. Theoretically, $p_1$ can either $p_r$ or $p_t$, so does $p_r$. We added more explanation in the refined version.

---

> ### Author Response · Authors · 2024-11-22
> **Feedback 2 to reviewer #zARm**
>
> **Q7**: Proposition 2 contains an over-general statement.
>
> **A7**: We really appreciate this point. We indeed did not provide rigorous proof of the proposition 2. We got this statement by jointly 1. investigating Matrix (7), especially the condition it has to satisfy to give the one and only one warping field~(as you mentioned the matrix rank), 2. practically run experiments to calculate the warping field by placing two receivers at two arbitrary positions in a 3D room scene at least two sound source speakers are emitting sound (where we experimentally show the nonexistence of the warping field). We are following your guidance to refine this section text description. Again, we really appreciate your guidance.
>
> **Q8**: Line 294 - Clarify how the Bilinear interpolation is performed.
>
> **A8**: Assume inside a scene with width $n$ and length $m$, the receiver's coordinate is $(x, y), 0 \leq x \leq n, 0 \leq y \leq m$. The learnable grid feature partition the room into $N \times M$ equally sized grids. Each grid is associated with a $D$-dimensional feature. To extract the receiver's position feature $v_{xy} \in \mathbb{R}^D$ from the grid feature, we perform bilinear interpolation on the grid feature. Specifically, for each of the D dimension, we use bilinear interpolation to extract the corresponding feature values at the position $(x, y)$. We then concatenate these feature values across all $D$ dimensions to form the $D$-dimensional position feature $v_{xy}$ for the receiver's position. This interpolation method is applied to extract both the target and the reference receiver position features.
>
> **Q9**: Motivation for using transformer.
>
> **A9**: Thanks for asking this question. In fact, we have tried numerous solutions before deciding to use the Transformer architecture. For example, we have tried,
>
> 1. NeRF [1] motivated MLP (multi-layer perception) to encode the warping field, in which the MLP neural network takes two positions (after either position encoding or full-connection to embed position to high dimension) as input and outputs a feature. The learned feature is added with warping field frequency index position encoding to predict the warping field one-by-one. It turns out predict all-zero warping field.
>
> 2. EMD (empirical mode decomposition) based warping field prediction. The motivation is decompose the highly irregular warping field into a set of simple and smooth warping fields, and predict each relatively smooth warping field separately. Again, we found the neural network (also MLP based) can just predict over-smooth warping field, and still fail at complex warping field.
>
> 3. To tackle the irregularity of the warping field, we also proposed to predict each value by combining raw localization (via classification, which initially localize the value within a raw range) and fine-grained small offset prediction (via regress head). It turned out to predict all-zero warping field.
>
> 4. We also tried several latest time-series prediction neural network, and found they couldn't handle the irregularity property of the warping field.
>
> We gradually finalize to adopt Transformer architecture after lots of prior attempts. Our assumption on why Transformer can predict reasonably good warping field includes: 1. using tokens to represent the warping field allow to implicitly capture the complex long-range dependency along the frequency axis, which is naturally achieved by multi-head self-attention layer. 2. asking each single token to predict a localized and non-overlapping warping field enables to neural network to predict a localized warping field in a relatively independent manner, and they were not confused or affected by global predictions.
>
> [1] Ben Mildenhall et al., NeRF: Representing Scenes as Neural Radiance Fields for View Synthesis, ECCV 2020.
>
> **Q10**: The proposed method is not necessarily more realistic compared to baselines. The required data is similar to collecting massive RIR data.
>
> **A10**: First, our proposed method isn't more realistic than RIR-based baseline like NAF in terms of training data size. This is partially because receiver-to-receiver warping field is much more irregular and non-smooth than RIR data (most RIR data is close to zero), training a reasonably good model naturally requires large training data. Second, our proposed method is more realistic in data collection. This is because SPEAR just requires two robots holding a receiver respectively to recording the reverberant sound which is relatively easy to deploy. However, RIR data collection is very tedious and difficulty to collect in real scenario, especially when we require humans to physically collect the RIR data. Third, we want to emphasize that our focus in this paper to predict a receiver position's spatial acoustic effects from another receiver by learning a warping field, it is intrinsically different in problem formulation and the way we think about this room acoustic effects prediction than RIR-based methods.

---

> ### Author Response · Authors · 2024-11-24
> **Feedback 3 to reviewer #zARm**
>
> **Q11 and Q15**: Not shown performance for NN
>
> **A11 and A15**: The NN baseline simply performs a nearest neighbour search, which has poor performance as shown in Table 1. Including the NN baseline performance makes it difficult to visualize the difference between other methods’ performance, so we didn’t include it.
>
>
> **Q12**: The figures order does not follow the order they appear in the text - Fig. 6 should come before Fig. 5.
>
> **A12**: Thanks for pointing this out. We have corrected this order issue in the revised version.
>
> **Q13**: Is the noise added during training? What happens in higher noise levels?
>
> **A13**: Yes, we have added noise interference during training (see Fig. 5 A and L 470-474), we can observe that while all three comparing methods (NAF, LinInterp, SPEAR) have seen performance drop (increased MSE metric) as more noise is involved, SPEAR maintains as the best-performing method and still outperforms the other two methods (NAF and LinInterp) by a large margin under all noise interference.
>
> **Q14**: Reverberation time and signals length.
>
> **A14**: The average RT60 is 0.19s for the Synthetic Dataset, 0.17s for the Photo-Realistic dataset, and 0.38s for the MeshRIR dataset split used in our experiment. We cut the source signal length to 1 second and used sampling rate of 16384. We will add these implementation detail in the refined version.
>
> **Q16**: Appendix D is empty.
>
> **A16**: We are really sorry for that, we have added the corresponding material in Appendix D in the revised version.
>
> **Q17**: Are there any insights regarding the characteristics of the failure cases?
>
> **A17**: First, as explained in L455-464, SPEAR struggles more to predict accurate warping field values in the high-frequency regions than in low-frequency regions. We assume this might be because high-frequency components correspond to the fine-grained details in the receiver-to-receiver warping field. More in-depth investigation is needed in improving the prediction accuracy in higher-frequency regions. Second, we experimentally found failure cases happen more frequently in geometric more complex 3D rooms (like Replica and Real world MeshRIR data), and areas close to barriers or walls. As the warping field becomes more irregular (or non-smooth) in these areas, meticulous designed framework in addressing these challenges awaits to be formulated.
>
> **Q18**: Reason for selecting 384 feature dimension.
>
> **A18**: We select the token dimension following the BERT model architecture, The BERT model used 768 token dimension. In order to reduce parameter size, we halved the dimension to 384 in our architecture. The number 43 is the token number, it is the number of tokens required to represent the 16384 length warping field (16384 / 384 = 42.67). Since 43 tokens of 384 dimension result in 16512 warping value predicted, the last 128 warping values are pruned to give 16384 warping values.

---

> > ### Author Response · Authors · 2024-11-29
> > **Thanks for Reviewing our Paper**
> >
> > Dear Reviewer zARm,
> >
> > We sincerely appreciate you spend time reviewing our paper and providing constructive feedback. During the rebuttal period, we have tried our best to provide feedback regarding your concerns. If you have any further concern, welcome to set up further discuss. We will be more than happy to provide more feedback on them.
> >
> > Best,
> > Authors

---

> > > ### Comment · Reviewer_zARm · 2024-11-30
> > > **Thanks for your response**
> > >
> > > Thank you for your detailed response and clarifications.
> > >
> > > Regarding your example (in A1), the data collected and the trained model will only be relevant for a specific source position. Additionally, it is required to have a dense set of recordings across the scene, which is not always feasible (a few tourists is not enough). Moreover, the method can only work for a single source which is not typical for choirs or orchestra. I believe that in this case it makes more sense to learn the RIR for different source and receiver locations so we can accommodate changes for both locations, enabling flexible usage in different future scenarios. Note that from two RIRs associated with a specific source location and each receiver location, we can compute the wrapping field between receivers.
> > >
> > > Overall, I find the current paper to be overly similar to existing works on relative transfer function estimation and RIR estimation, particularly DeepNeRAP. Additionally, the practicality of learning a wrapping field corresponding to a fixed source position remains unconvincing to me.

---

### Official Review · Reviewer_CYyz · 2024-11-01

**Soundness:** 2
**Presentation:** 3
**Contribution:** 2
**Rating:** 5
**Confidence:** 3

**Summary:**

The authors propose a novel method for estimating the warping field that corresponds to sound propagation between two points inside a fixed environment. The method applies for a stationary source and two moving receivers (microphones). The receivers are synchronized, and thus the transfer function between the reference to the target receiver can be estimated from recordings in the frequency domain.

**Strengths:**

* The method can be applied relatively easily since it does not require a lot of knowledge about the environment
  * It is relatively original since most other methods require more information about the environment or a more complex recording setup
  * Quality is somewhat unclear (see comments below)
  * Clarity and significance could be improved (see comments below)

**Weaknesses:**

* The method estimates the ratio between two transfer functions and this is prone to ill conditioning - however, it is not fully addressed. In the experiments they simply use clipping and zeroing to handle such cases, but how this affects performance is not clear
* The strengths and weaknesses of the method are not sufficiently clear
* I’m finding some difficulties in understanding exactly what SPEAR is trying to learn. In line 304: “...we can obtain the ground truth warping field…” - if you can calculate it analytically and then train the model to predict it, what exactly does the model learn? Some kind of generalization to other frequencies? Interpolation to grid points that are not measured? In any case, this is not discussed in the experiments
* Experiments section (section 4.6)
  * There is too much emphasis on visual analysis of the warping field signal in the frequency domain (Figures 4, 6, 7, 8). It is very difficult
to understand actual performance from these graphs. In all cases it looks like the estimated warping field is very different from the ground truth
  * Fig 5 - what is the meaning of the MSE values? Again, difficult to understand something about performance from reading absolute MSE values (i.e., how bad/good is an MSE of 1.06?)
  * Table 3 - was the metric measured for the same signal? It should be noted that depending on the frequency content of the signal, estimating the warping field is limited
* It is not clear how to use the method given some very important parameters:
  * How close should the receiver be? Is this frequency dependent?
  * What source signal to use?
  *How many grid points should be sampled in the environment in order to estimate the warping field of the entire environment? Is this even possible?

**Minor comments**
* Introduction → contribution 3: “We demonstrate SPEAR superiority…” - superiority compared to what?
* Lines 132-139: the relation to “Time-series Prediction” is not very clear. Please explain more clearly what is the relation between SPEAR and the type of networks you mentioned.
* Equation (2) - is p_1 and p_2 the same as p_r and p_t ? if so, please be consistent with notation
* Lines 308-309: “For ground truth warping field supervision, we combine both L1 and L2 loss.” - can you please provide an explicit equation for the loss? Is there a weighting parameter between the L1 and L2 losses? Why did you incorporate both L1 and L2?

**Questions:**

* SPEAR learns the propagation of sound from one receiver (reference) to the second receiver (target). So to map all potential (reference, target) positions in the room is also combinatorically complex (as in the RIR case). Thus:
  * How much more efficient this method is to the previous ones (RIR-based / NAF)?
  * Please explain more clearly the benefits (or tradeoffs) between SPEAR and source-to-receiver modeling methods. In other words, why does the latter require prior space acoustic properties knowledge and SPEAR does not? (this is not clear in both the introduction and not in “related work” section)
  * How dependent is the method on the specific source signal? If the signal is narrowband (single frequency), then the method would not be able to estimate other frequencies.
* There are two very significant claims in lines 162-165:
  * Warping transform is multiplicative in the frequency domain
  * The acoustic neural warping field is independent on audio content
While you refer to the appendix for proof and discussion, I would suggest adding some coarse and intuitive explanation to why that is the case.
* Equation (5) - there are cases where the transfer function H may be zero for some frequencies. Then the relations in equation (5) would not hold. How do you handle these cases? (in the experiments you mention that you use a Sine Sweep signal but this is not clear at this stage of the paper)
* Equation (7) is a generalization of eq. (5): thus the determinant condition applies to equation (5) as well, but it is not mentioned.
* In your SPEAR training you fix the position of the source. If we would like to use SPEAR with a source that is located in a different position (and in a different room), we would have to train it again in the new setting, correct? If so, this should be mentioned clearly
* What applications could benefit from this method (especially compared to alternatives)?

---

> ### Author Response · Authors · 2024-11-22
> **Feedback 1 to reviewer #CYyz**
>
> We sincerely appreciate the reviewer's constructive feedback and detailed comments of our work. We provide detailed feedback regarding all the concerns raised by the reviewer.
>
> **Weakness part**
>
> **Q1**: The method estimates the ratio between two transfer functions and this is prone to ill conditioning - however, it is not fully addressed. In the experiments they simply use clipping and zeroing to handle such cases, but how this affects performance is not clear.
>
> **A1**: Thanks for pointing this out. As you said, estimating the ratio between two transfer functions inevitably leads to abnormal value like NaN and exceedingly large values. We adopt "clipping'' (within $[-10,10]$) and "zeroing'' to handle those anomaly. We have had detailed evaluation on the impact of clipping/zeroing process and find those two operations just change less than 3\% data points, and they lead to very subtle human perception difference and quantitative evaluation result difference. For example, we heard two recovered audios by clipping the warping field $[-10,10]$ and $[-100,100]$, respectively, and couldn't perceive any difference between them. In terms of quantitative metrics like PSEQ and SSIM, they still give small variation that is within $0.01$. We have added these details in the refined version.
>
> **Q2**: The strengths and weaknesses of the method are not sufficiently clear.
>
> **A2**: Thanks for mentioning this. We listed the strength of our method in the Introduction section last paragraph (line 92-99), and weakness in the section 5 (line 537-539). We highlight them here:
>
> *Strengths*: First, SPEAR doesn't require prior space acoustic properties knowledge such as geometric layout and construction material acoustic properties. Second, it was designed to be agnostic to sound source, and can be efficiently trained with receivers' collected data. Third, it can be trained with more readily accessible data -- by just requiring two receivers to collected sound at discrete positions.
>
> *Weaknesses*: The first weakness is dense sampling which prevents us from training SPEAR with less data. The second weakness is that we assume the two data collection receivers are time-synchronized so that they can collect the same sound content at discrete positions, this requirement may introduce extra synchronization effort when deploy receiver-robots to collect the data.
>
> **Q3**: If you can calculate the warping field analytically and then train the model to predict it, what exactly does SPEAR learn?
>
> **A3**: Given a room scene, we collect data at discrete position pairs (so we just have ground truth warping field for discrete position pairs) and aim to train a continuous warping field SPEAR that is capable of predicting warping field relating two arbitrary positions that are not collected in the training dataset. We presented this in Introduction section line 76-84 and Fig. 1.
>
> **Q4**: There is too much emphasis on visual analysis.
>
> **A4**: In the main paper, we can only use figures to directly visualize the warping field, in the supplementary material, we provide the warping field effected audio waveform for more intuitive comparison. We agree that our difference exists between prediction and ground, but SPEAR result remains as the most similar to the ground truth in terms of the overall warping field pattern.
>
> **Q5**: Fig 5 - what is the meaning of the MSE values?
>
> **A5**: We calculate the MSE loss by considering the predicted the warping field as a long vector (16384 in our experiment configuration), and calculate the MSE between the predicted vector and the ground truth warping field vector. We agree that it is difficult to ground a MSE value to a physical meaning of the prediction quality, but its simplicity and lack of normalization make it a reliable measure of the predicted warping field's quality, independent of the ground truth warping field values. Therefore, it is included as a metric to compare the performance between different learning methods.
>
> **Q6**: Table 3 - was the metric measured for the same signal?
>
> **A6**: The metric reported in Table 3 are test performance of our model for different sound sources when trained on the sine sweep signal, as explained in section Audio-Content Agnostic Verification (line 510). Truly, our model requires that the training source signal frequency range covers the frequency range of the testing sound source, and this is addressed in line 303.

---

> > ### Comment · Reviewer_CYyz · 2024-11-28
> > **Response to Feedback 1**
> >
> > A1: thanks for clarifying. However, this still not solved the main issue of not being able to estimate the warping field at points where one of the transfer functions H1 or H2 is zero. There are many methods to regularize this ill conditioned expressions, and it is a very common practice to perform that. I still think that this issue should be addressed. At least, how often does ill conditioning occurs in your experiments?
> >
> > A2: I think another weakness should be added, which is: What are the constrains on the received locations relative to one another in order for the method to work? If you do not control the receiver locations, how prone is the method for bad warping field estimations?
> >
> > A3: But what is still unclear to me, is how well can you extrapolate to new room positions? If you interpolate between two very close measured locations that is one thing. But what happened if you only sampled 50% of the room positions? Then, how do you know that you have sampled the room sufficiently?
> >
> > A4: My concern is that there are better evaluation methods than simply observing the signal in the time domain. I could not really learn or understand the performance of the method by looking at these signals.
> >
> > A5: Similar to above. If this was a normalized MSE, then very small values (like -20 dB) would mean that estimation is quite good. But with current MSE values, its difficult to understand.

---

> > > ### Author Response · Authors · 2024-11-29
> > > **Further Feedback 1 to Reviewer #CYyz**
> > >
> > > **Additional Evaluation Result**
> > >
> > > Thank you for your suggestion on further evaluation metric. We have re-evaluated our model and the baseline methods' performance on generated audio quality using normalized MSE dB. The evaluation datasets are the same as those used for evaluation in the main paper. The results are shown below:
> > >
> > > | normalized MSE (dB) | Synthetic       | Photo-Realistic       | MeshRIR      |
> > > |--------|--------------|--------------|--------------|
> > > | Interp | 1.88  | 1.81   | 1.63  |
> > > | Nneigh | 6.25  | 4.92  | 5.68  |
> > > | NAF    | -0.76 | -0.82 | -1.34 |
> > > | SPEAR  | -1.69 | -1.58 | -1.70 |
> > >
> > > Our model outperforms all baseline methods across all datasets. However, under this normalized MSE dB level, human perception system can still tell the difference between the ground truth and the generated reverberant audio sample. We are exploring other model architectures and training methods in order to improve the model performance

---

> > > > ### Comment · Reviewer_CYyz · 2024-12-01
> > > > **Response to "Further Feedback 1 to Reviewer #CYyz"**
> > > >
> > > > Thanks for incorporating normalized MSE. As I suspected, these are very large errors, such that other than comparing the method to other there is no real use in these analysis. And as stated previously, the evaluation study heavily relies on analyzing the signal in the time domain. In order to provide more insights into the quality of the method, other evaluation criteria are needed.

---

> ### Author Response · Authors · 2024-11-22
> **Feedback 2 to reviewer #CYyz**
>
> **Q7**: How close should the receiver be? Is receiver position frequency dependent? What source signal to use? Is the receiver sampling density practical?
>
> **A7**: The receiver position sampling strategy is explained in section 4.2 data sampling strategies (line 347). The receivers are arranged on a grid for Synthetic and MeshRIR dataset, and randomly positioned in Photo Realistic dataset. The average distance between receivers are 0.05 meters. We did not find correlation between the receiver sampling density and the frequency of the sound source.
>
> Our method requires the sound source used in training to cover the frequency range of the sound in inference. For simplicity, we use the full frequency range sine-sweep sound as the sound source. This is briefly mentioned in line 303 and explained in detail in the revised version. As explained in line 319, we sampled 3000 audios in the Synthetic dataset, and presented the model’s performance in ablation experiment (line 515) when the sampling density decreases. We agree that this sampling density limits the methods’ ability to be deployed to real world applications, as noted in the conclusion of our paper. We are exploring strategies such as leveraging advanced learning methodologies and incorporating spatial smoothness constraints to reduce the dependency on dense sampling.
>
>
> **Q8**: Relation to Time-series Prediction models.
>
> **A8**: Our method's output is similar to that of the time-series prediction models, since the warping field in frequency domain can be treated as a time-series. However, as we show in line 308-309, warping field prediction exhibits no causality, but instead is only dependent on two receiver positions.
>
> **Q9**: Detail about L1 and L2 loss.
>
> **A9**: Please see the revised version for the explicit loss equations. We selected L2 loss to achieve higher convergence speed at the start of the training. To further improve model performance at frequencies where the predicted warping field L2 loss is less than 1, we added the L1 loss in the loss function. We took the inspiration from prior works [1, 2] to use the combined L1 and L2 loss. No relative weighting parameter between the L1 and L2 losses is used.
>
> [1] Hui Zou et al., Regularization and Variable Selection via the Elastic Net
>
> [2] Nathan Howard et al., A Neural Acoustic Echo Canceller Optimized Using An Automatic Speech Recognizer And Large Scale Synthetic Data, ICASSL 2021.
>
>
> **Questions part**
>
> **Q1**: How much more efficient this method is to the previous ones (RIR-based / NAF)?
>
> **A1**: Thanks for pointing this out. First, the problem formulation of SPEAR is different from RIR-based methods like NAF. SPEAR's motivation is to get rid of source-to-receiver RIR estimation, but instead learn receiver-to-receiver warping field estimation without knowing the source position. Second, in experiment, we compared with NAF, and show better performance over NAF across all evaluation metrics. Third, we also compared with other RIR-based methods including Fast-RIR [1] and TS-RIR [2] by tweaking their input and retraining these method to predict the receiver-to-receiver warping field. We found these two methods predict all-zero warping field.
>
> [1] Ratnarajah, A et al., Fast-RIR: Fast Neural Diffuse Room Impulse Response Generator, ICASSP, 2022.
>
> [2] Ratnarajah, A et al., TS-RIR: Trans-lated Synthetic Room Impulse Responses for Speech Augmentation. ASRU, 2021.
>
> **Q2**: When the transfer function H may be zero for some frequencies, the relations in equation (5) would not hold. How does the training method handle 0 values in transfer function H?
>
> **A2**: Thank you for identifying this. Indeed, when the Fourier transform of RIR is zero at certain frequencies, the equation (5) does not hold. When values in the denominators of equation (5) is zero or are close to zero, the ground truth warping field corresponding to the frequency will be NAN or a significantly large value. We handle these cases by filling NAN values in the ground truth warping field with 0, and clip the warping field in range $[-10,10]$. We found that there tends to be only a little number of frequencies are modified after the preprocessing ($<100$ out off 16384 frequencies), and the preprocessed ground truth warping field does not result in significant difference in the generated audio result (Figure V in the appendix).
>
> **Q3**: Eq. (7) is a generalization of Eq. (5)
>
> **A3**: Thank you for your insight. Eq. (5) is a specific condition of Eq. (7), and writing Eq. (5) in Eq. (7)'s form is straightforward. In Eq. (5) the element-wise division between $H_2$ and $H_1$ can be expressed as matrix multiplication between $diag(H_2)$ and $diag(H_1)^{-1}$. Thus, assuming there is no zero values in the frequency transformation $H_1$, the inverse of the diagonal matrix $diag(H_1)$ is simply another diagonal matrix with the elements $diag(H_1)^{-1}(ii)$ on the diagonal being $\frac{1}{H_1(i)}$.

---

> ### Author Response · Authors · 2024-11-22
> **Feedback 3 to reviewer #CYyz**
>
> **Q4**: Using SPEAR with a source that is located in a different position.
>
> **A4**: Indeed, if users wish to render acoustic effect of a sound source at different position, they need to train a new SPEAR model. We briefly mentioned it in the problem formulation (line 144), and will explicitly address the problem in the conclusion section in the revised version.
>
> **Q5**: What application could benefit from the proposed method.
>
> **A5**: Our model is effective in the following scenario: Suppose a few tourists uploaded their recorded audio of one scene, (e.g. audio taken in a concert or speech given in a chapel), and one other user wants to explore the scene online using the recorded audio. As this online user virtually moves in the scene, our model predicts the warping field that changes the audio from the ones recorded in known positions (uploaded by the tourists) to the audio heard at the position of the virtual user. Our end-to-end method is capable of rendering the spatial audio heard by the online user without needing to know the position and the content of the sound source.

---

> ### Author Response · Authors · 2024-11-24
> **Feedback 4 to reviewer #CYyz**
>
> **Q6**: Benefits (or trade-offs) between SPEAR and source-to-receiver modelling methods.
>
> **A6**: Thanks for pointing this out. First, the underlying difference between SPEAR and source-to-receiver methods (or RIR-based methods) lies in way they choose to predict spatial acoustic effects. SPEAR adopts receiver-to-receiver way, while source-to-receiver modelling methods go through source-to-receiver way. Second, **SPEAR Advantage**: SPEAR framework doesn't require source information, it is source-agnostic. Moreover, SPEAR requires training data that is more readily accessible. It does not require RIR data that is exceedingly difficult to collect in real scenario. Instead, it just requires two receivers to record the spatial sound at various discrete position, which is much easier to deploy in real scenario. Third, **SPEAR Disadvantage**: given the irregularity property of receiver-to-receiver warping field (Sec. 3.4), currently SPEAR require densely sampled data to train a reasonably good model. We proposed several potential solutions for mitigate this issue. In summary, SPEAR is a totally different methodology in modelling spatial acoustic effects than source-to-receiver modelling methods, it shows preferable advantages over source-to-receiver methods.

---

> ### Author Response · Authors · 2024-11-28
> **Thanks for Reviewing our paper**
>
> Dear Reviewer CYyz,
>
> We sincerely appreciate your efforts in providing highly professional reviews. During the rebuttal period, we have tried our best to provide feedback, discussion regarding your concern. All of them will be reflected in our final revised version.
>
> If you have any further concern, welcome to set up further discuss. We will be more than happy to provide more feedback on them.
>
> Thank you again for your constructive comment.
>
> Best,
>
> Authors

---

> ### Comment · Reviewer_CYyz · 2024-11-28
> **Response to Feedback 3**
>
> A4: thanks
>
> A5: OK, so the method is useful for spatial reproduction of a recorded environment. What I wanted to understand is: in what cases does SPEAR has a benefit over other methods in the literature? In order to reproduce the spatial sound in a given room, SPEAR requires training on this specific room. What is then the use-cases compared to other methods?

---

> ### Comment · Reviewer_CYyz · 2024-11-28
> **Response to Feedback 4**
>
> A6: Thanks for clarifying. Please make it clearer in the revised version. One caveat is that there are RIR-based methods that are also source agnostic.

---

### Official Review · Reviewer_y6xA · 2024-11-04

**Soundness:** 3
**Presentation:** 3
**Contribution:** 3
**Rating:** 6
**Confidence:** 3

**Summary:**

The paper introduces SPEAR, a novel neural warping field model designed to predict spatial acoustic effects in a 3D environment with a single stationary audio source. Unlike traditional source-to-receiver models requiring prior knowledge of room acoustics, SPEAR operates from a receiver-to-receiver perspective, allowing it to predict how audio would sound at different spatial positions using only discrete audio recordings at various receiver positions. This framework is trained using synthetic, photo-realistic, and real-world datasets, demonstrating significant flexibility and generalizability across different environments. The paper's contributions include a new problem formulation, a theoretically supported neural architecture guided by three physical principles, and comprehensive experimentation showing SPEAR's accuracy and efficiency over baseline methods.

**Strengths:**

1. The receiver-to-receiver formulation of spatial acoustics is innovative, providing a new paradigm in spatial audio modeling that does not rely on prior knowledge of acoustic properties.
2. Methodologically rigorous, supported by a blend of theoretical analysis and robust experimental results across diverse datasets (synthetic, photo-realistic, real-world).
3. The overall flow and structure of the paper are clear, with detailed explanations of each stage in the model’s development, including physical principles and architecture specifics.
4. SPEAR’s method addresses a gap in the field, making spatial acoustics modeling more accessible for real-world applications in complex environments.

**Weaknesses:**

1. Sampling Density Requirement: A dense sampling of receiver positions is currently required for SPEAR to achieve optimal accuracy. This requirement may limit its scalability in highly variable environments.

2. Positioning Constraint: SPEAR assumes all receiver positions lie on the same horizontal plane, which could restrict applications in multi-level or irregular environments. Addressing this limitation would extend the model’s utility.

**Questions:**

1. Could the authors elaborate on potential methods to reduce the dense sampling requirement? Would techniques like data augmentation be feasible for this purpose?

2. Are there specific changes or enhancements that could allow SPEAR to handle multi-level environments with varying elevations?

3. For real-time applications, what optimizations could be implemented to further reduce inference time without sacrificing accuracy?

4. Could the authors provide a comparison with conventional RIR-based approaches, detailing the trade-offs in accuracy, efficiency, and data requirements?

---

> ### Author Response · Authors · 2024-11-22
> **Feedback 1 to Reviewer #y6xA**
>
> Thank you for your positive feedback on the originality and technical quality of our work.
>
> **Q1:** The dense sampling requirement may limit its scalability in highly variable environments.
>
> **A1:** We thank the reviewer for raising an important point regarding the sampling density requirement of receiver positions in our SPEAR framework. We understand that the need for dense sampling could appear as a limitation in terms of scalability for diverse environments. We wish to provide further clarification on why this requirement is necessary and how it aligns with the underlying design and goals of our model:
>
> 1. **Sensitivity to Position Changes**: As detailed in Sections 3.4 and 4.7 of our paper, the receiver-to-receiver warping field is highly sensitive to changes in receiver positions. Even small shifts can lead to significant variations in the predicted warping field due to the complex nature of spatial audio propagation, which includes interactions such as reflection, diffraction, and absorption. This position sensitivity, visualized in Figure 2 of the paper, underscores the importance of dense sampling to ensure that the model accurately captures these intricate spatial acoustic effects.
>
> 2. **Ensuring Accurate Warping Field Prediction**: Dense sampling helps mitigate the risk of inaccuracies when predicting the warping field for target positions not encountered during training. Our experiments, illustrated in Table 4, empirically demonstrate that as the sampling density decreases, the model's performance significantly drops, resulting in higher MSE and lower SSIM. This is because the learned warping field must interpolate between known positions to predict novel target positions accurately, a task made more feasible with closer-spaced training data.
>
> 3. **Core Challenge and Future Work**: We acknowledge that this dense sampling requirement can be challenging for highly variable environments. However, it is a necessary step given the current sensitivity of the warping field to position changes. Addressing this challenge and improving model efficiency with fewer sampled positions is an active area for future research, as noted in the conclusion of our paper. We are exploring strategies such as leveraging advanced learning methodologies and incorporating spatial smoothness constraints to reduce the dependency on dense sampling.
>
> We hope these points provide a clearer understanding of why dense sampling is crucial for SPEAR's current implementation and how it supports the accuracy and robustness of our predictions.
>
> **Q2**: Addressing the limitation of all receiver positions lie on the same horizontal plane would extend the model’s utility.
>
> **A2**: We thank the reviewer for pointing out this point. The reason why we put all receivers lie on the same horizontal plane is two-fold: First, in real-scenarios, most receivers remain at the same height at various positions across the flat room floor, such as we human ear and robot carrying a receiver at the fixed height. Second, as we mentioned in the previous question about the sampling density and section 3.4 in the main paper, the receiver-to-receiver neural warping field is highly position-sensitive, a subtle position change will lead to significant warping field change (see Fig. 2 for the visual and warping field change comparison). That is, allowing the receivers' position height change inevitably introduces extra data due to the dense sampling strategy. We are running another experience in which we relax the receivers height to be changeable, and we will report the result once it comes out.
>
> **Q3**: Are there specific changes or enhancements that could allow SPEAR to handle multi-level environments with varying elevations?
>
> **A3**: Thanks for pointing this out. As we presented in Q2, SPEAR is naturally capable of handling multi-level environments with varying elevations because SPEAR accepts two receivers position and can implicitly encode the multi-level environment acoustic properties relating to warping field learning.
>
> **Q4**: For real-time applications, what optimizations could be implemented to further reduce inference time without sacrificing accuracy?
>
> **A4**: Currently, our proposed SPEAR neural network just has parameter size 27 M, so the latency would not be a big issue.
>
> For real-time applications, we can further 1. quantize the neural network from float32 to unsigned int8 (fine-tuning might further be needed to control accuracy reduction), or 2. adopt teacher-student curriculum learn to train a large SPEAR model and further distil the knowledge of the large SPEAR to another small SPEAR model so that the final optimized small SPEAR model can achieve real-time inference without sacrificing accuracy.

---

> ### Author Response · Authors · 2024-11-22
> **Feedback 2 to Reviewer #y6xA**
>
> **Q5**: Could the authors elaborate on potential methods to reduce the dense sampling requirement? Would techniques like data augmentation be feasible for this purpose?
>
> **A5**: We sincerely thank the reviewer for pointing out this vital challenge. We also noted that dense sampling strategy used in this paper is the key challenge that deserves future investigation. We are currently working on designing novel solutions to reduce the sampling density. Here are some of our initial ideas to reduce the sampling density:
>
> 1. Pre-train a sound propagation aware foundation model that essentially captures the general underlying sound propagation related characteristics (aka room acoustic parameters). Given the scarcity of receiver-to-receiver warping field data, we can alternatively pre-train a foundation SPEAR model on spatial audio simulated in a scene with similar size, and fine-tune on the receiver-to-receiver warping field data to learn the warping field specific to the target scene layout. Our expectation is that fine-tuning on top of the pre-trained sound propagation aware foundation model to learn receiver-to-receiver warping field would require less-dense sampling data and give better result at the same time.
>
> 2. Treat warping field prediction as audio generation task. Since the receiver-to-receiver warping field basically is a time-series, we can equivalently treat it as an audio waveform and the warping field prediction then becomes audio generation problem. We can then use generative model such as latent diffusion model (LDM [1]) and vocoder (e.g. HiFiGAN [2]) to predict the warping field. The dense sampling might be mitigated by the generative model's generalizability.
>
> 3. Position-shift tolerance training. Given the high irregularity of warping field (warping field values in adjacent frequency bins have large variant values), we find it easily leads to all-zero warping field prediction (silent warping field) if we apply L1 loss (or the combination of L1 and L2 loss) to each frequency separately under low sampling density of receiver position. One mitigate training strategy is to incorporate position-shit tolerance training strategy, in which the core idea is to relax the per-frequency-independent prediction to allow a frequency's warping field value to be predicted by its neighbouring frequencies. For example, we won't introduce penalty (loss) if a frequency's warping field value is successfully predicted by its neighbourhood. Such training strategy might encourage the whole neural network to learn representative warping field and further reduce the sampling density.
>
> [1] Robin Rombach et al., High-Resolution Image Synthesis with Latent Diffusion Models, CVPR 2022.
>
> [2] Jungil Kong et al., HiFi-GAN: Generative Adversarial Networks for Efficient and High Fidelity Speech Synthesis, NeurIPS 2020.
>
> **Q6**: Could the authors provide a comparison with conventional RIR-based approaches, detailing the trade-offs in accuracy, efficiency, and data requirements?
>
> **A6**: Thanks for pointing this out. In the main paper, we have compared with NAF [3], which is a RIR based approach and we show we can achieve much better performance than NAF. We have also tried other relevant RIR based methods, such as Fast-RIR [4] and TS-RIR [5] all of them led to all-zero warping field prediction.
>
> [3] Luo, A. et al., Learning Neural Acoustic Fields, NeurIPS, 2022.
>
> [4] Ratnarajah, A et al., Fast-RIR: Fast Neural Diffuse Room Impulse Response Generator, ICASSP, 2022.
>
> [5] Ratnarajah, A et al., TS-RIR: Trans-lated Synthetic Room Impulse Responses for Speech Augmentation. ASRU, 2021.

---

> ### Author Response · Authors · 2024-11-25
> **Feedback 3 to Reviewer #y6xA**
>
> **Experiment on Relax the Receivers Height to be Changeable**
>
> We performed experiment to verify our model could still perform with variable receiver height. We modified our model to use a 3 dimensional grid, and performed bilinear interpolation with 3D coordinates of the target and reference receiver. Apart from this, no modification is done on the model structure. We simulated using the same scene geometry, sound source position, and sound source content as the Synthetic dataset used in our experiment. We placed 12800 receivers on a 3D grid with spatial resolution $0.05m$, covering a $4m \times 2m \times 0.2m$ space. 11000 receivers are used for training, and the rest 1800 receivers are used for testing. The model performance is shown below:
>
> | Method    | SDR   | MSE  | PSNR  | SSIM | PESQ |
> |-----------|-------|------|-------|------|------|
> | LinInterp | -0.58 | 1.52 | 13.27 | 0.87 | 1.48 |
> | NNeigh    | -2.81 | 2.53 | 13.89 | 0.71 | 1.25 |
> | SPEAR     |  **1.13** | **1.03** | **14.83** | **0.89** | **1.63** |
>
> The PSNR and SSIM of warping field predicted by our model is not significantly affected by the additional axis introduced. On the contrary, the SDR and MSE of the predicted warping field, and the PESQ of the generated reverberant speech all improved over the proposed model trained on 2D receiver positions.
>
> We believe this is because introducing an additional dimension increases the number of training receiver microphone neigbouring to the testing receivers. The neighbouring receivers of receiver $A$ refer to those that are directly adjacent to $A$ along the x, y, or z axes. In 2D space, a testing receiver will have maximum 4 neighbouring receivers from the training dataset, while in 3D space, the testing receiver can have at most 6 neighbouring receivers from the training dataset. Learning from more neighbouring receivers' warping fields could leads to the improvement in performance. In conclusion, this shows our learning paradigm's capability to perform with receivers positioned at different height.

---

### Author Response · Authors · 2024-11-25
**Feedback to All Reviewers**

We appreciate all reviewers for the constructive feedback and suggestions to our work. All of them are useful for us to continue to improve our work.

Based on the reviewers' suggestion and concerns, we have revised the paper and updated a new version of paper. In revised version, we have corrected all typos and made necessary clarifications. We labelled all revisions in blue color.

We have provided separate feedback to each reviewer. We further address two main concerns raised by reviewers here.

1. SPEAR versus RIR-based methods.

Although both SPEAR and RIR-based methods tend to model spatial acoustic effect for a given receiver position, their underlying motivations are totally different. While RIR-based methods go through source-to-receiver pipeline to model the room impulse response, SPEAR is completely source-agnostic and warps the acoustic effects perceived by one receiver position to the other receiver position, so it goes through receiver-to-receiver pipeline which hasn't been formally discussed previously. **Our main focus in this paper is to provide an alternative solution to model spatial acoustic effects, which is in contrast with existing source-to-receiver (or RIR-based) methods. Our target isn't to show RIR-based methods are inferior to SPEAR.**

In terms of the data acquisition, SPEAR enjoys two advantages. First, it doesn't require the audio source position and we assume sometimes getting to know the audio source position is a challenging task. Second, SPEAR requires neither RIR data which is very inefficient and challenging to collect in real scenarios (although the advantage would be damped since we require the audio source emits sine-sweep audio during training phase), nor room acoustic characteristics (like geometric layout, furniture deployment, etc.). It just requires two robots (or humans) holding a microphone to record the acoustic environment.

2. Train SPEAR with Sine-sweep audio but test SPEAR with arbitrary sound.

We agree that requiring sine-sweep audio to train SPEAR will inevitably reduce SPEAR's practicability in real scenario. Since we want to learn a general warping field that can handle arbitrary sound in the room, we need to use to the sound signal covering the full-frequency. It is theoretically feasible to train SPEAR with more common sound such as machine, animal and human sound, but the learned SPEAR will collapse into local frequency that is only suitable to handle the sound used in training. We show the capability of sine-sweep trained SPEAR in handling a variety of common sounds in Table 3 and ablation section 4.7 during testing phase. Using special audio to acoustically perceive an environment is essential in order to achieve more general and specific-sound independent perception.

---

### Meta-Review · Area_Chair_K9oE · 2024-12-17

**Metareview:**

This paper received contrasting reviews, indeed, 3 out of 4 reviews leaned more to negative evaluations (rating 5), while 1 was extremely positive (max rating, 10). After rebuttal and extensive discussion, the 2 of the former evaluations downgraded the rating to clearly negative (rating 3), one remained on 5, while the latter rating was decreased to slightly positive (rating 6).

This result was obtained after extensive discussion between the authors and the reviewers, and also the positive reviewer (rev. y6xA) significantly re-evaluated his/her judgment after the comments and discussion raised by the other reviewers.

In conclusion, for the above reasons, this paper cannot be considered acceptable to ICLR 2025.


The work deals with the estimation of the spatial acoustic effects in a 3D space for any receiver position when a stationary source emits sound. Not requiring neither source position nor 3D space acoustic properties, the proposed method SPEAR simply uses two microphones to actively record the spatial audio independently at discrete positions, and produces in output a warping field warping the recorded audio on reference position to target position and comparing it with real recorder audio. In short, SPEAR aims at acoustically characterizing the 3D space from a receiver-to-receiver perspective.

Major good points relate to the originality of the work and the fact of not using much information for the task it addresses (differently from former works).
Weak points are several, regarding the clarity of the methodology in some parts, unclear contributions of the work, weak demonstration of good performance and several issues concerning the experimental results, not fully clear motivations, the need of better comparison with the state of the art.

The authors discussed extensively wth reviewers, who acknowledged to have clarified some of these issues, but in the end they were not fully satisfied of the answer authors provided.

**Additional Comments On Reviewer Discussion:**

Reviewer y6xA after an initial high (max) scoring revised his/her evaluation downgrading the rating after the rebuttal and the consequent discussions, agreeing on some of comments raised by the other reviewers.

---

### Decision · Program_Chairs · 2025-01-22

Reject